# No Need for Ad-hoc Substitutes: The Expected Cost is a Principled All-purpose Classification Metric

**Luciana Ferrer**                                                          *lferrer@dc.uba.ar*
*Instituto de Ciencias de la Computación*
*CONICET - Universidad de Buenos Aires, Argentina*

**Reviewed on OpenReview:** *https://openreview.net/forum?id=5PPbvCExZs*

## Abstract

The expected cost (EC) is one of the main classification metrics introduced in statistical and machine learning books. It is based on the assumption that, for a given application of interest, each decision made by the system has a corresponding cost which depends on the true class of the sample. An evaluation metric can then be defined by taking the expectation of the cost over the data. Two special cases of the EC are widely used in the machine learning literature: the error rate (one minus the accuracy) and the balanced error rate (one minus the balanced accuracy or unweighted average recall). Other instances of the EC can be useful for applications in which some types of errors are more severe than others, or when the prior probabilities of the classes differ between the evaluation data and the use-case scenario. Surprisingly, the general form for the EC is rarely used in the machine learning literature. Instead, alternative ad-hoc metrics like the F-beta score and the Matthews correlation coefficient (MCC) are used for many applications. In this work, we argue that the EC is superior to these alternative metrics, being more general, interpretable, and adaptable to any application scenario. We provide both theoretically-motivated discussions as well as examples to illustrate the behavior of the different metrics.

## 1 Introduction

During the development of machine learning systems, the model architecture, hyperparameters, training approaches, and other system characteristics, are determined based on the performance measured on a validation set. After development, the performance of the resulting system is reported using a separate evaluation set. In both of these steps it is essential that the metric used to evaluate performance be reflective of the needs of the application of interest. In this work, we focus on the study and comparison of metrics that are designed to evaluate classification systems that produce a single categorical label per sample based only on the information extracted from that sample. While systems trained for this task can sometimes be used for other purposes, like ranking of samples or producing an interpretable score of each class, the metrics needed to evaluate such systems are out of the scope of this work.[1]

One of the most widely used metrics for the evaluation of categorical decisions is the total error rate or, equivalently, the accuracy, which is equal to one minus the total error rate. These metrics assume that all errors are equally costly regardless of the true class of the sample. In some applications, though, like those involving medical (Ashby & Smith, 2000; Kornak & Lu, 2011), biometric (Van Leeuwen & Brümmer, 2007; Gonzalez-Rodriguez, 2014), or business and economic decisions (Harsanyi, 1978; Berger, 2013), the costs may be extremely different across error types. In those cases, alternative ad-hoc metrics like the F-beta score (Rijsbergen, 1979) or the Matthews correlation coefficient (MCC) (Matthews, 1975), are often used for evaluation. In this work, we argue that the expected cost (EC), a decades-old generalization of the error rate

---

[1]Ranking systems that eventually produce categorical decisions by labelling the top M samples as being from the target class are also outside of the scope of this paper, since the resulting label for each sample is not based only on that sample but on the scores for other samples under evaluation.

(Savage, 1972), can be successfully used in all classification scenarios where categorical decisions are made and where the severity of an error depends only on the true class of the sample and the decision made.[2] Other metrics designed for this scenario, like the F-beta score or the MCC can be seen as ad-hoc substitutes for the EC, offering no advantages and various disadvantages over this metric.

The EC is based on a simple general assumption: given a certain use-case scenario a numerical value can be assigned to each combination of true class $h$ and decision $d$, reflecting the cost incurred by the user of the system when making decision $d$ for a sample of class $h$ (Elkan, 2001; Berger, 2013; Savage, 1972). Once these costs are defined for the application of interest, the performance of a classification system can be obtained as the expectation of the cost over the data. This process results in the EC metric, which is also sometimes called expected prediction error (Hastie et al., 2001), or expected loss (Bishop, 2006), and is equivalent to the negative expected utility (DeGroot, 1970; Bernardo & Smith, 1994). As various authors have argued, optimizing the EC is the principled way to make rational decisions (Good, 1952; Peterson, 2009; Dyrland et al., 2022; Russell & Norvig, 2010). Note that, while the EC definition is very simple, the set of costs it relies upon are sometimes difficult to define given a certain application of interest (Kahneman, 2011; Dyrland et al., 2022). Yet, as we discuss in this work, costs are also implicitly involved in all alternative classification metrics. The benefit of the EC is that the costs are selected explicitly rather than being obscured within the metric's definition.

The EC is used in statistical learning as the loss to be minimized when making decisions based on posterior probabilities using Bayes decision theory (Hastie et al., 2001; Bishop, 2006). Yet, this metric can be used to evaluate classification systems regardless of how decisions are made (Hernández-Orallo et al., 2012) and, hence, it is not constrained to the evaluation of systems that produce posterior probabilities. The error rate and the balanced error rates which correspond to one minus the accuracy and balanced accuracy (or unweighted average recall), respectively, are, in fact, special cases of the EC which are widely used in the literature to assess performance of categorical decisions. Further, the general form of the EC with unequal costs has been widely adopted by the speech processing community as an evaluation metric for the tasks of speaker verification and language recognition (see, e.g., Brümmer, 2010; Brümmer et al., 2021; Van Leeuwen & Brümmer, 2007; Greenberg et al., 2020), where it is called detection cost function (DCF). The DCF is the metric of choice for the periodic evaluations organized by NIST.[3] Other than for these tasks, though, we are not aware of other applications for which the general form of the EC is used as standard practice. While some papers stand out proposing the use of EC for medical tasks (Ashby & Smith, 2000; Kornak & Lu, 2011; Godau et al., 2023), they are exceptions in a literature where various other metrics are used instead (Hicks et al., 2022). Interestingly, the EC has been recently highlighted as a "so far uncommon" metric with many desirable features, in a recent paper on metrics for medical image analysis written by an international consortium of experts (Maier-Hein et al., 2024) .

In this work, we review and compare the EC with other more widely-used classification metrics in the machine learning literature. Our study is related to a recent work by Dyrland et al. (2022) who similarly argued in favor of the use of a decision theoretical metric (expected utility, in their case) and against ad-hoc metrics like the F-beta and the MCC, showing empirical evidence on synthetic data that these alternative metrics do not correspond to an expected utility. In this work, we provide a thorough analysis of the EC which complements the work by Dyrland et al. (2022) in various aspects. First, we explain how the EC can be used under various different scenarios, including the case of unequal error severity, the case where the decisions do not coincide with the classes (e.g., when the system has an "abstain from choosing a class" option), and the case where the class priors in the evaluation data differ from the one expected in practice. We propose to address the latter case, quite common in some disciplines, by allowing the class priors to be parameters of the metric instead of being defined by the class frequencies in the evaluation data, as in all standard classification metrics. This allows us to manipulate the priors when needed, setting them to the ones that are expected at deployment. With this generalization of the EC, it is also possible to see the balanced error rate as a special case of the EC obtained when the costs depend on the class priors. We also introduce the normalized EC (NEC), a more interpretable version of the EC obtained by dividing the EC by the EC of the

---

[2]In some scenarios, the cost of an error depends on characteristics other than just the true class of the sample (e.g., Zadrozny & Elkan, 2001), or are determined by the end user after deployment. Such scenarios are out of the scope of this paper.

[3]http://www.nist.gov/itl/iad/mig/sre.cfm

best naive system. Finally, we compare the EC with alternative classification metrics, like the F-beta score and the MCC, both empirically, like Dyrland et al., though with alternative visualizations and including results in real datasets, but also theoretically by finding the relationship between the expressions. Based on this comparison, we argue that the EC is superior to those metrics, being more intuitive and flexible, having a theoretical framework for making optimal decisions, and having useful properties that facilitate its interpretation. The code used to compute all metrics and generate all plots and tables in this paper is available at `https://github.com/luferrer/expected_cost`. We hope this work will encourage the machine learning community to embrace this valuable but underused metric.

## 2 The expected cost (EC)

The EC is defined as a generalization of the probability of error for cases in which errors cannot be considered to have equally-severe consequences. In the context of decision theory, the EC is used as the function to be minimized in order to make optimal decisions when the system outputs posterior probabilities. See, for example, the work by Bishop (2006, Section 1.5), Hastie et al. (2001, Section 2.4), Duda et al. (2001, Section 2.2), Elkan (2001), DeGroot (1970), Dyrland et al. (2022), and Savage (1972). Yet, EC can be used to assess performance of any classification system, regardless of how categorical decisions are made or whether they make such decisions based on posterior probabilities or some other type of score.

The EC is the expectation of a cost function which assigns a numerical penalty to each combination of the true class of the sample, $h$, and decision made by the system, $d$. Since we assume that both $h$ and $d$ are categorical, this cost function can be specified by a matrix with components $C(h, d)$, which represent the cost we believe that the system should incur for making decision $d$ when the true class of the samples was $h$.

While there is no restriction on the values assigned to the costs, for every matrix with entries $C(h, d)$, a new matrix equivalent to the original one can be created with entries given by $C'(h, d) = C(h, d) - \min_d C(h, d)$. The equivalence is in the sense that systems would be ranked in the same way by both costs matrices (DeGroot, 1970; Russell & Norvig, 2010; Brümmer, 2010). The resulting costs are non-negative and the best decision for each class has a cost of zero, which often facilitates the analyses. In the rest of this paper, we assume that all costs matrices have been standardized in this way.

In this work, as generally done in the machine learning literature, the expectation is computed with respect to the empirical distribution in the test data, i.e., as an average of the cost over the test samples. Given a test set $\{(h_1, d_1), \ldots, (h_N, d_N)\}$, where $h_t \in \mathcal{H} = \{H_1, \ldots, H_K\}$ are the true labels and $d_t \in \mathcal{D} = \{D_1, \ldots, D_M\}$ are the system's decisions for sample $t$, the EC is given by:

$$\text{EC} \quad = \quad \frac{1}{N} \sum_{t=1}^{N} C(h_t, d_t) \tag{1}$$

where $C(h_t, d_t)$ is the cost incurred on sample $t$ with true class $h_t$ and system's decision $d_t$. This expression can be rewritten as:

$$\text{EC} \quad = \quad \frac{1}{N} \sum_{i=1}^{K} \sum_{j=1}^{M} c_{ij} N_{ij} \quad = \quad \sum_{i=1}^{K} \sum_{j=1}^{M} c_{ij} P_i R_{ij} \tag{2}$$

where $c_{ij} = C(H_i, D_j)$, $R_{ij} = N_{ij}/N_{i*}$ is the fraction of samples from class $H_i$ for which the system made decision $D_j$, with $N_{i*} = \sum_j N_{ij}$ being the number of samples of class $H_i$ in the test set, and where $P_i = N_{i*}/N$ is the empirical estimate of the prior probability of class $H_i$ in the evaluation data. The first expression in this equation is derived by using the fact that the cost is the same for each combination of class $H_i$ and decision $D_j$. Hence, the summation over the samples in Equation (1) can be converted to a (double) summation over every combination of $i$ and $j$ by multiplying the corresponding cost by the number of samples for that combination of class and decision, $N_{ij}$.

The second expression for the EC in Equation (2) allows for a useful generalization. While $P_i$ is originally given by $N_{i*}/N$, it can potentially be changed arbitrarily. This allows us to set the priors so that they coincide with those we expect to see when the system is deployed which, in many applications, do not necessarily

coincide with those observed in the evaluation data. Rather than forcing the evaluation data to have the priors we expect to see in practice, which would imply downsampling or upsampling the data, we can simply use those priors when computing the cost and use the evaluation data as-is to compute the $R_{ij}$ values. This advantage of the EC has been highlighted by Godau et al. (2023), where the authors show the benefits of using an EC metric with the priors determined based on the deployment data.

Further, note that in the definition of the EC the set of possible decisions do not need to coincide with the set of classes. Rather than being the predicted class labels, the decisions may be the actions that could be taken based on the information in the input sample (Duda et al., 2001, Section 1.3.12). Hence, the EC allows for the evaluation of more general classification systems than other classification metrics like the F-beta score or the MCC, which assume that the set of decisions corresponds to the set of possible class labels. Consider, for example, a (simplified) medical scenario where a doctor needs to make a decision based on an MRI of the brain of a patient that may or may not have a tumor. The decisions made by the doctor do not necessarily need to be one of the two classes (patient does/does not have a tumor). They could instead be the actions that the doctor could take based on the MRI image which could be, for example: "perform surgery", "send home", "do more tests". A cost for each combination of true class and decision can then be selected based on the severity of each type of error. Internally, the system can make decisions in order to optimize the resulting EC, as explained in Section 2.4. The "do more test" decision would be taken in cases in which the system was not certain enough about the class of the sample to make a decision to operate or send the patient home.

## 2.1 Normalized expected cost

As explained above, any cost or utility matrix can be mapped to an equivalent cost matrix with $c_{ij} \geq 0$, and at least one $c_{ij} = 0$ for each class $H_i$. The minimum value for any EC with these properties is 0, attained for a perfect system that always chooses a decision with cost 0 for each sample. The maximum value that EC can take, though, depends on the costs and priors used to compute it. This makes it hard to assess whether a certain value of EC is good or bad. For this reason, it is convenient to compute a normalized version of the EC (NEC) given by:

$$\text{NEC} = \frac{\text{EC}}{\text{EC}_{\text{naive}}} \quad \text{where} \quad \text{EC}_{\text{naive}} = \min_{\hat{\jmath}} \sum_{i=1}^{K} c_{i\,\hat{\jmath}}\, P_i \tag{3}$$

The value used for normalization is the EC for the best naive system that does not have access to the input sample and, hence, always makes the same decision $\hat{\jmath}$. The system is free to choose $\hat{\jmath}$ to optimize the EC for the chosen costs and priors. The expression for $\text{EC}_{\text{naive}}$ can be easily derived by considering that, for a system that always makes the same decision $\hat{\jmath}$, $R_{ij}$ is 0 for all $j \neq \hat{\jmath}$ and 1 for $j = \hat{\jmath}$. Plugging those values of $R_{ij}$ in Equation (2) we find that the EC for that system is $\sum_{i=1}^{K} c_{i\,\hat{\jmath}}\, P_i$. The minimum over all possible decisions $\hat{\jmath}$ is the EC of the best naive system.

For the special case where there are two possible true labels ($K = 2$) and two possible decisions ($M = 2$), and we assume that $c_{11} = c_{22} = 0$, the NEC reduces to:

$$\text{NEC} \quad = \quad \frac{c_{12}P_1 R_{12} + c_{21}P_2 R_{21}}{\min(c_{12}P_1, c_{21}P_2)} \quad = \quad \begin{cases} \alpha\, R_{12} + R_{21} & \text{if } \alpha \geq 1 \\ R_{12} + \alpha^{-1}\, R_{21} & \text{otherwise} \end{cases} \quad \text{where} \quad \alpha = \frac{c_{12}P_1}{c_{21}P_2} \tag{4}$$

Hence, in this scenario, there is only one free parameter to choose in the NEC. All combinations of priors and costs that lead to the same $\alpha$ correspond to the same metric.

The NEC metric has an essential property: While its value can be larger than 1, this only happens for systems that are worse than a naive system. In particular, a system with NEC larger than 1 can be trivially improved upon by replacing it with one that always makes the same least-costly decision $\hat{\jmath}$. Yet, discarding the system is usually not necessary. For systems that first generate a score for each class and then make decision based on those scores, a NEC larger than 1 indicates that the decision stage was not properly optimized. In the binary case, optimization of the decision stage can be achieved by simply tuning the decision threshold. The problem of optimizing the decision stage for a certain EC of choice is discussed in Section 2.4.

## 2.2 Selection of costs

The selection of costs for each type of error is not necessarily trivial as it requires an understanding of the application scenario and, hence, should be done in consultation with field experts. This issue is discussed at length, both in general and for specific families of applications, for example, by Raiffa (1970b); Russell & Norvig (2010); Kahneman (2011); Hunink et al. (2014); Dyrland et al. (2022). Below, we give a brief overview of the subject including some general guidelines. We refer the reader to those references for an in-depth discussion.

When determining the cost matrix, it is often useful to allow for a combination of positive and negative costs (a negative cost would be equivalent to a utility) or to define, instead, utility matrices, where bad decisions correspond to negative values and good decisions to positive ones. Utility matrices can be converted into cost matrices by multiplying them by minus one. Further, it is also useful to note that shifting each row by a constant or globally scaling cost matrices does not affect the optimal decisions or the ranking of systems (Brümmer, 2010, Section 3.4). In particular, this means that, for the standard binary case with one decision corresponding to each class, all cost matrices can be parameterized by a single value. Equation (4) shows that this parameter is determined by the ratio between the costs for the two types of error.

Costs or utilities can often be defined in monetary terms. This may be the case for companies using the output of machine learning systems to make business decisions. For example, consider a simplified loan approval scenario. A bank may use an automated system to predict whether a person will be able to repay their loan and, based on its output, make one of three possible decisions: deny the loan (D), approve the loan with a standard interest rate (AS), or approve the loan with a higher rate (AH). The utility for a person that will pay off the full loan will be 0 for decision D (assuming the cost of processing the application is negligible), and a positive value for AS and AH given by the total amount of interest that the person will pay in each case, which we will express as $I_S = \alpha_S L$ and $I_H = \alpha_H L$, where $L$ is the principal loan amount requested and the $\alpha$s depend on the corresponding interest rate and the payment schedule. We further need to reduce the utility for the AH case by some factor $\beta$ given by the fraction of clients that would decide to accept the loan despite the higher interest rate offered. We assume that all clients offered the lower interest rate accept the loan. For a person that will default on their loan, the utility for decision D would again be 0, while the utility for decisions AH and AS would be the fraction of the loan that was paid minus the total amount loaned. Assuming that, on average, a fraction $\gamma$ of the total money owed (principal plus interest) is paid by borrowers before defaulting, the utility for those cases may be computed as $\gamma(L + I_H) - L$ and $\gamma(L + I_S) - L$. Note that these utilities may be negative. Here we assume that borrowers that end up defaulting would always accept the high-interest loans since other banks would also offer them high rates. Hence, in this simplified loan approval scenario, the utility matrix would be given by:

$$U = \quad L \begin{array}{c} \\ \end{array} \begin{bmatrix} \overset{D}{0} & \overset{AH}{\beta\,\alpha_H} & \overset{AS}{\alpha_S} \\ 0 & \gamma(1 + \alpha_H) - 1 & \gamma(1 + \alpha_S) - 1 \end{bmatrix} \begin{array}{l} \text{repays the full loan and the interest} \\ \text{does not repay the full loan} \end{array}$$

where the principal value of the loan, $L$, was factorized outside of the matrix. As mentioned above, once defined in this way for convenience, we can multiply the matrix by -1 to turn it into a cost matrix $C$ and normalize it by doing $C(h, d) - \min_d C(h, d)$ so that the minimum value in each row is 0.

As illustrated by this example, when the only possible outcomes of the decisions made with the system are monetary, selecting the cost matrix is relatively straight forward, though perhaps not trivial: the costs or utilities simply reflect the losses or gains incurred by each decision for a given class, expressed in some currency. In general, though, decisions may have consequences that are not only monetary. For example, consider an application for finding a given singer within a large database of songs. In this case, as mentioned above, only the ratio of costs for the two error types needs to be determined. We need to decide how costly it is to not find a song that includes the target singer, usually called a *miss*, compared to selecting a song that does not include it, usually called a *false alarm*. If we wish to simply retrieve some examples of the singer, false alarms may be more costly than misses, with the ratio of costs depending on how bothered we believe the user would be by being offered a song that does not contain the singer of interest. If we wish to find every single audio that includes that singer – for example, for copyright reasons – we should assign a much higher cost to misses than to false alarms. In this case, perhaps we are back in a scenario where we can

specify the costs in monetary terms since misses imply a loss of revenue for the artist and false alarms need to be filtered by human annotators that require compensation for every reviewed song.

In some applications, like those related to medical decisions, the selection of costs may involve setting a value for a human life. While this is, undoubtedly, a very uncomfortable decision to make, refusing to set a value for a human life may implicitly result in its being undervalued. Russell & Norvig (2010) give a compelling example involving a government agency that commissioned a study on removing asbestos from schools: "The decision analysts performing the study assumed a particular dollar value for the life of a school-age child, and argued that the rational choice under that assumption was to remove the asbestos. The agency, morally outraged at the idea of setting the value of a life, rejected the report out of hand. It then decided against asbestos removal—implicitly asserting a lower value for the life of a child than that assigned by the analysts."

Sometimes the determination of exact costs is not possible but we have some knowledge about the range of values they should take. In this case, costs may be specified as a (discrete or continuous) distribution. Interestingly, the expectation of the cost when the individual costs are random variables reduces to the standard EC above, with costs given by their expected values (Dyrland et al., 2022; Raiffa, 1970a). Further, in some scenarios, one might wish to select systems based on a trade off between two or more ECs, as well as other systems characteristics like memory consumption or run time.

While the difficulty in selecting costs may seem a disadvantage of the EC, it is important to note that the trade-off between different types of error is encoded in every classification metric (this issue is discussed in Sections 3 and 4). The advantage of the EC over other metrics is that the trade-off is explicit and transparent. Hence, instead of letting the metric choose the trade-off for us, we can make our best effort in choosing costs that are, at least approximately, relevant for our task of interest. As illustrated by the asbestos removal example mentioned above, refusing to explicitly select the costs may result in the implicit selection of extremely incorrect ones.

### 2.3  Error rate and accuracy

The standard error rate (also called total error or probability of error) is given by:

$$\text{ER} \quad = \quad \frac{1}{N} \sum_{i=1}^{K} \sum_{j=1|j\neq i}^{K} N_{ij} \tag{5}$$

Comparing this equation with Equation (2) we can see that the ER is a particular case of EC obtained when $M = K$, and $c_{ij} = 1$ for $i \neq j$ and $c_{ij} = 0$ for $i = j$, which we will call the "0-1 cost matrix". Further, since the accuracy is given by one minus the ER, we can see that accuracy is also just a trivial function of one specific case of EC.

In the ER computation, errors from all classes are weighted equally. Hence, when the classes are highly imbalanced, the ER and the accuracy become somewhat insensitive to the performance in the minority classes. For this reason, when errors in the detection of the minority classes are considered more severe than those in the majority classes, the balanced ER (BER), also known as weighted ER, is used as metric instead of the ER. Alternatively, the balanced accuracy, given by one minus the balanced ER, sometimes called unweighted average recall (UAR), is also commonly used in these scenarios. The BER is defined as the average of the ER values per class. That is:

$$\text{BER} \quad = \quad \frac{1}{K} \sum_{i=1}^{K} \frac{1}{N_{i*}} \sum_{j=1|j\neq i}^{K} N_{ij} \quad = \quad \frac{1}{K} \sum_{i=1}^{K} \sum_{j=1|j\neq i}^{K} R_{ij}. \tag{6}$$

This expression coincides with the EC when $M = K$ and the costs are selected such that $c_{ij} = N/(KN_{i*}) = 1/(KP_i)$ for $i \neq j$ and $c_{ij} = 0$ for $i = j$. As a consequence of setting the costs this way, all error rates ($R_{ij}$ for $i \neq j$) have the same influence on the final metric. Note that by deriving the balanced error rate as a special case of the EC we can see exactly what is being assumed when we compute this metric: that the cost for each type of error is given by the inverse of the frequency of the true class of the sample.

The general form of EC offers greater flexibility than these two special cases of the standard and balanced error rates, allowing us to accommodate scenarios in which the errors are neither all equally costly nor as costly as the inverse of the prior for each class. The EC allows us to think of each type of error independently and set their cost according to their severity.

## 2.4 Optimizing the decisions

Most modern classifiers are composed of two stages, one that generates posterior probabilities for the classes and a second one that makes the categorical decisions based on those posteriors. The EC and NEC can be used to assess the quality of decisions made with any strategy, like the threshold tuning approach commonly used for the two-class setting. Yet, a unique advantage of these metrics is that the optimal decisions for systems that produce posterior probabilities can also be obtained in closed form. The decisions that theoretically optimize the EC are called Bayes decisions (Hastie et al., 2001; Bishop, 2006; Peterson, 2009). Given an EC determined by a specific cost matrix $c_{ij}$, the Bayes decision for a sample $x$ is given by:

$$d_B(x) = \arg\min_{\hat{j}} \sum_{i=1}^{K} c_{i\hat{j}} \, P(H_i|x) \tag{7}$$

where $P(H_i|x)$ is the posterior probability for class $H_i$ given sample $x$ as produced in an internal stage of our classifier.

One important condition needs to be satisfied for Bayes decisions to be optimal: the posteriors used to make the decisions need to be well calibrated. Loosely speaking, for well-calibrated posteriors, 90% of the times the posterior for a certain class $h$ is 0.9, the true class is indeed $h$. For a formal definition of calibration, please refer, for example, to the works by DeGroot & Fienberg (1983); Brümmer (2010); Filho et al. (2023). Unfortunately, good calibration is not always achieved by modern classifiers mostly due to overfitting of the parameters to the training data, which results in overconfident posteriors (Guo et al., 2017). Fortunately, though, calibration quality can usually be easily fixed, if needed, by doing post-hoc calibration on the output of the classifier (Filho et al., 2023; Ferrer & Ramos, 2025). The topic of calibration is out of the scope of this paper, but it is covered at length in the publications cited above and many others. The repository provided with this work includes tools for assessing and fixing calibration.

Assuming our classifier is calibrated, either inherently so or after adding a post-hoc calibration stage, the EC provides us with an elegant approach for making optimal decisions, even for the multi-class case. In fact, as discussed in Appendix A, the argmax decisions so widely used in the machine learning literature correspond to the Bayes decisions for the EC defined with the 0-1 cost matrix, i.e., the error rate (notably, this method is generally used without verifying whether the calibration condition is satisfied). For the binary case with one decision per class, Bayes decisions can be made by comparing one of the two posteriors with a threshold which depends on the costs (see Appendix A.2). In the general case, making Bayes decisions for an EC of choice is as simple as implementing Equation (7). In Section 4, we show examples on making Bayes decisions for various EC metrics.

To summarize, the EC offers an important advantage over other classification metrics in that, when well-calibrated posterior probabilities are available, decisions can be made in a principled way using Bayes decision theory. Alternatively, if well-calibrated posteriors are not available, decisions can be made with any of the approaches used for other metrics, like threshold tuning.

## 2.5 Optimizing the classifier

The section above describes how to optimize the decisions for an EC of interest. Those decisions are made using the posteriors output by a classifier. Notably, even when systems are developed for a specific EC of interest, the classifiers are not usually trained to optimize such EC. In fact, perhaps the most common procedure in the machine learning literature on classification is to train a classifier to optimize cross-entropy while assessing its performance using accuracy (which, as mentioned above, is one minus the EC for the 0-1 cost matrix). The reason why the classifiers are not trained to optimize accuracy is that this metric, as any

EC and any other metric computed on categorical decisions, is not differentiable with respect to the scores, making its use as objective function for model training inconvenient.

In general, training classifiers that will be evaluated in terms of an EC using a cross-entropy objective is a very reasonable approach. To see this, we can refer to an interesting theorem that shows that cross-entropy can be written as an integral over a family of ECs computed over Bayes decisions (Brümmer, 2010, Section 8.2). By optimizing cross-entropy we are optimizing a trade-off between a family of ECs. A model trained with cross-entropy is encouraged to produce good posterior probabilities over the whole simplex, including the operating point corresponding to accuracy and infinitely many others. In some cases, though, we might not want to settle for a model that optimizes a trade-off between ECs, most of which are of no interest for our application. In such cases, we may wish to directly train the model to optimize our EC of interest.

In those scenarios, a relaxation of the EC can be used to turn it into a differentiable function that approximates the original metric, enabling its use as an objective function during optimization. The use of the EC for the purpose of model optimization is outside of the scope of this paper where we focus on their use for system evaluation. Yet, for completeness, here we mention an approach for using the EC as training objective proposed by Mingote et al. (2019) for a binary problem with one decision per class. First, note that, for this case, the error rates in Equation (2) can be written as

$$R_{12} = \frac{N_{12}}{N_{1*}} = \frac{1}{N_{1*}} \sum_{t=1|h_t=H_1}^{N} I(P(H_1|x_t) < u) \tag{8}$$

$$R_{21} = \frac{N_{21}}{N_{2*}} = \frac{1}{N_{2*}} \sum_{t=1|h_t=H_2}^{N} I(P(H_1|x_t) > u) \tag{9}$$

where $I$ is the indicator function, which is 1 when its argument is true and 0 otherwise. In these equations, $N_{12}$ and $N_{21}$ are the number of samples for which the wrong decision was made because the posterior fell on the wrong side of the threshold $u$.

The indicator function is what makes the EC not differentiable. Mingote et al. (2019) proposed to approximate the indicator function by a sigmoid, computing the rates as:

$$R_{12} = \frac{1}{N_{1*}} \sum_{t=1|h_t=H_1}^{N} \sigma(u - P(H_1|x_t)) \tag{10}$$

$$R_{21} = \frac{1}{N_{2*}} \sum_{t=1|h_t=H_2}^{N} \sigma(P(H_1|x_t) - u) \tag{11}$$

Plugging those rates into Equation (2) we obtain a differentiable version of the EC. In their approach, the threshold $u$ is optimized along with the rest of the parameters of the model.

This simple relaxation of the EC allows for its use as objective function during training. Unfortunately, generalizing this approach to the multi-class case is not trivial since the decision regions cannot be obtained by simple thresholding. We are not aware of any work that proposed a differentiable relaxation of the EC for the multi-class case.

## 3 Other classification metrics

In this section we describe some of the most common classification metrics in the machine learning literature and compare them to the EC metric. We also describe the area under the ROC curve (AUC) and the equal error rate (EER). These metrics are not strictly classification metrics in the sense considered in this work, since they do not evaluate the quality of categorical decisions but of the scores that would eventually be used to make such decisions after choosing one specific threshold. Yet, we include them in this section for completeness since they are widely used in the machine learning classification literature.

### 3.1 F-beta score

The F-beta score, here $F_\beta$, is defined as one minus a metric called effectiveness, which was first introduced by Rijsbergen (1979). The $F_\beta$ is probably one of the most widely used metrics for binary classification in modern machine learning literature, alongside accuracy and error rate. For example, in the SuperGLUE benchmark[4] of natural language processing (NLP) tasks, out of 10 tasks, 3 use the $F_\beta$ (one uses MCC and the rest use accuracy). The SQuAD benchmark[5] for question-answering has two metrics, one of which is $F_\beta$. The IEMOCAP benchmark[6] for emotion recognition is evaluated with (weighted, micro or macro) $F_\beta$ as well as accuracy. The slot-filling SLURP benchmark[7] for spoken language understanding uses $F_\beta$. The $F_\beta$ is also widely used in medical imaging, where it is called Dice coefficient (Taha & Hanbury, 2015), in opinion mining (Wang et al., 2019), and for epileptic seizure detection (Siddiqui et al., 2020). These are just a few examples from a long list of tasks where the $F_\beta$ is selected for evaluation of performance, highlighting the wide spread use of this metric.

The $F_\beta$ assumes that there are only two classes, that the set of decisions coincides with the set of classes ($K = M = 2$ and $\mathcal{D} = \mathcal{H}$), and that the problem is not symmetric; one of the classes is taken as the class of interest to be detected. Here, we will take the class of interest to be class $H_2$. $F_\beta$ is defined as follows:

$$F_\beta \;\; = \;\; (1+\beta^2)\, \frac{\text{Precision Recall}}{\beta^2\,\text{Precision} + \text{Recall}} \quad \text{where} \quad \begin{aligned} \text{Precision} &= N_{22}/N_{*2} \\ \text{Recall} &= N_{22}/N_{2*} \end{aligned} \tag{12}$$

Replacing those values for the precision and recall, we get:

$$F_\beta \;\; = \;\; \frac{(1+\beta^2)N_{22}}{(1+\beta^2)N_{22} + \beta^2 N_{21} + N_{12}} \tag{13}$$

$F_\beta$ takes values between 0 and 1. Larger values indicate better performance, contrary to the EC for which larger values indicate worse performance. For this reason, in order to compare $F_\beta$ with EC, it is convenient to work with $1 - F_\beta$, which is given by:

$$1 - F_\beta \;\; = \;\; \frac{\beta^2 N_{21} + N_{12}}{(1+\beta^2)N_{22} + \beta^2 N_{21} + N_{12}} \;\; = \;\; \frac{\beta^2 R_{21}P_2 + R_{12}P_1}{\beta^2 P_2 + R_{*2}} \tag{14}$$

where $R_{*2} = N_{*2}/N$ is the fraction of samples in the evaluation set that are labelled as class $H_2$. The numerator in this expression is the EC for the binary case when costs for mistakes are given by $c_{12} = 1$ and $c_{21} = \beta^2$, which we will call $EC_{\beta^2}$:

$$EC_{\beta^2} \;\; = \;\; \beta^2 R_{21}P_2 + R_{12}P_1. \tag{15}$$

We can now express $1 - F_\beta$ as a function of $EC_{\beta^2}$ and its normalized version:

$$1 - F_\beta \;\; = \;\; \frac{EC_{\beta^2}}{\beta^2 P_2 + R_{*2}} \;\; = \;\; \min(\beta^2 P_2, P_1)\frac{NEC_{\beta^2}}{\beta^2 P_2 + R_{*2}} \tag{16}$$

That is, $1 - F_\beta$ is proportional to $EC_{\beta^2}$ with a scaling factor given by the inverse of $\beta^2 P_2 + R_{*2}$. The relationship between $F_\beta$ and $EC_{\beta^2}$ was shown through simulations by Dyrland et al. (2022, Figures 2 and 3), who also discuss the inadequacy of $F_\beta$ as a classification metric. Below, we analyze Equation (16) theoretically, sheding light into the issues that make $F_\beta$ a poor classification metric.

The first term in the denominator of Equation (16), $\beta^2 P_2$, is independent of the system. When comparing different systems on the same dataset (or on different datasets with the same class priors) $P_2$ is fixed.[8] The

---

[4]https://super.gluebenchmark.com/leaderboard

[5]https://rajpurkar.github.io/SQuAD-explorer/

[6]https://paperswithcode.com/sota/emotion-recognition-in-conversation-on

[7]https://paperswithcode.com/sota/slot-filling-on-slurp

[8]Note that values of $F_\beta$ across datasets with different priors are not comparable to each other. In particular, the $F_\beta$ computed on a certain evaluation dataset is not a good predictor of the $F_\beta$ that will be obtained during deployment if the priors between the two sets differ. This scenario can be addressed in a principled way with the EC by setting the priors in Equation (2) to those that are expected during deployment (Godau et al., 2023).

second term in the denominator in Equation (16), $R_{*2}$, is the percentage of samples that are labeled by the system as being of class $H_2$. Given two systems with the same value for $EC_{\beta^2}$, $F_\beta$ will favor the one that detects more samples as being from class $H_2$, regardless of whether they are correctly or incorrectly classified (examples of this behavior will be given in Section 4). This is an odd thing to reward: having more samples from class $H_2$ detected is not beneficial in itself. If, for the application of interest, detecting the samples from class $H_2$ is more important than detecting the samples of class $H_1$, the principled approach for defining a metric is to use an EC which penalizes errors in samples from class $H_2$ more than errors in samples of class $H_1$ (i.e., setting $c_{21} > c_{12}$).

As for the EC, we can compute the best $F_\beta$ that would correspond to a naive system that always outputs the same decision. If the system always chooses class $H_1$, $F_\beta = 0$ since $N_{22} = 0$. On the other hand, if the system always chooses class $H_2$, the recall is 1 and the precision is equal to the prior of class $H_2$, $P_2$, which results in $F_\beta = (1 + \beta^2)P_2/(\beta^2 P_2 + 1)$. This is the value of $F_\beta$ for the best naive system. Any system with an $F_\beta$ worse than this value should be considered ineffective. Since the $F_\beta$ for the best naive system depends on $P_2$, a given value of $F_\beta$ should be interpreted differently depending on the dataset. A value of $F_1$ of 0.4 is much better than chance for a dataset with $P_2 = 0.01$ which would have a naive $F_1$ of 0.02, but only just above chance for a dataset with $P_2 = 0.2$ for which the naive $F_1$ is 0.333. While it would be possible to normalize the $F_\beta$ to make it more readily interpretable, as far as we know, this is never done in the literature.

Finally, note that, unlike for the EC where, as long as the scores are calibrated, the optimal decisions can be made with Bayes decision theory, as far as we know there is no equivalent decision theory for the $F_\beta$. For the specific scenario for which the $F_\beta$ is defined, which corresponds to binary classification with one decision per class, decisions can be made by thresholding and the optimal threshold can be obtained empirically using development data. Note, though, that, even for perfectly calibrated scores, the optimal threshold for $F_\beta$ will depend on the system under evaluation through $R_{*2}$ rather than being fixed, as for the EC. Since every threshold corresponds to the best threshold for a certain EC, we can interpret an $F_\beta$ that is minimized at a specific threshold as being equivalent to the EC that is optimized at that same threshold: they are equivalent in the sense that both metrics reach their optimum value for the same threshold. Hence, if the optimal threshold for $F_\beta$ changes for each system, the implicit costs assigned to each type of error also change. This means that $F_\beta$ optimizes for a different trade off between error types depending on the system under evaluation. This is, in our view, an undesirable characteristic for a metric. We will see examples of this behavior in Section 4.1.1.

In summary, the $F_\beta$ has several issues that make it a suboptimal metric: 1) it is restricted to scenarios with two classes and two decisions, 2) the interpretation of its value depends on the priors of the dataset, 3) the effective costs being optimized depend on the system under evaluation rather than being determined by the needs of the application, and 4) it cannot be optimized theoretically. All of these problems are solved by the NEC metric.

## 3.2 Matthews correlation coefficient

The Matthews correlation coefficient (MCC) was first introduced by Matthews (1975) for comparison of chemical structures, later proposed as a metric for binary classification by Baldi et al. (2000), and finally generalized to the multi-class case by Gorodkin (2004). In this section we will consider the binary version since this is enough to show its flaws as classification metric. Chicco & Jurman (2020) argue that MCC is the most informative single score to establish the quality of a binary classifier that outputs hard decisions. EC is not considered as an option in that paper. As we discuss below, we believe EC is superior to MCC as a metric for classification performance.

As the $F_\beta$, the MCC assumes that the set of decisions is the same as the set of classes. For the binary case MCC is defined as follows:

$$\text{MCC} \quad = \quad \frac{N_{11}N_{22} - N_{12}N_{21}}{\sqrt{(N_{11} + N_{21})(N_{11} + N_{12})(N_{22} + N_{12})(N_{22} + N_{21})}} \tag{17}$$

After some manipulations and using $R_{ij} = N_{ij}/N_{i*}$, we get

$$\text{MCC} = \sqrt{\frac{N_{1*}N_{2*}}{N_{*1}N_{*2}}} \left(1 - (R_{12} + R_{21})\right) = \sqrt{\frac{P_2(1-P_2)}{R_{*2}(1-R_{*2})}} \left(1 - \text{NEC}_\text{b}\right) \tag{18}$$

where $\text{NEC}_\text{b} = R_{12} + R_{21}$ is the normalized BER (this can be seen by dividing Equation (6) by the BER of the naive systems which is 0.5 for the binary case). As for the $F_\beta$, the relationship between MCC and $\text{EC}_{\beta^2}$ was shown through simulations by Dyrland et al. (2022, Figures 2 and 3). Here, we analyze the relationship theoretically. We show empirical results in Section 4.

From Equation (18) we see that when the $\text{NEC}_\text{b}$ is larger than 1, which, as explained in Section 2.1, happens when the system is worse than the best naive system, the MCC is negative. Further, we can see that MCC and $1 - \text{NEC}_\text{b}$ differ by the square root factor. The numerator in this factor is fixed, it does not depend on the system, only on the priors (see footnote 8, which also applies to the MCC). The denominator, on the other hand, does depend on the system and is largest when $R_{*2} = 0.5$ and smaller as the decisions become more imbalanced (i.e., as $R_{*2}$ approaches 0 or 1). Hence, the factor will grow as the decisions become more imbalanced, making MCC larger in absolute value. For two systems with the same $\text{NEC}_\text{b} < 1$, MCC will favor the one with larger imbalance in the decisions. As discussed with respect to $F_\beta$, this behavior does not seem reasonable since the frequency with which a system makes a certain decision is not a good or bad quality in itself. As the $F_\beta$, the MCC effectively optimizes for a different trade off between error types depending on the system under evaluation. Finally, also as for $F_\beta$, as far as we know, the MCC does not have a corresponding decision theory and needs to be optimized empirically.

### 3.3 Net benefit

Net benefit, a metric used in binary classification for some medical applications (Vickers et al., 2016; Riley et al., 2021; Cowley et al., 2019), is defined as:

$$\text{NB} = \frac{N_{22}}{N} - \frac{p}{1-p}\frac{N_{12}}{N} \tag{19}$$

where $p$ is a parameter of the metric called threshold probability for reasons that are explained below. We can rewrite this expression as follows:

$$\text{NB} = P_2 R_{22} - \frac{p}{1-p}P_1 R_{12} = P_2 - \left(P_2 R_{21} + \frac{p}{1-p}P_1 R_{12}\right) \tag{20}$$

where we used the following equalities: $P_i R_{ij} = N_{ij}/N$ and $R_{22} + R_{21} = 1$. The term in parenthesis in this equation is the EC when the number of labels and decisions is 2 ($K = M = 2$) and $c_{21} = 1$, and $c_{12} = p/(1-p)$. We can then express NB as a function of the corresponding normalized EC, which we will call $\text{NEC}_p$:

$$\text{NB} = P_2 - \min\left(P_2, \frac{p}{1-p}P_1\right)\text{NEC}_p \tag{21}$$

Hence, while NB is an affine function of a NEC, making them equivalent for ranking systems, the NB looses the interpretability of the NEC given by the fact that a NEC of 1.0 indicates that the system has the same performance as a naive system. For this reason, we believe the normalized EC is preferable to the NB.

NB's single parameter $p$ is defined in such a way that it coincides with the optimal threshold for this metric. We can see this by first noting that the optimal threshold for NB is the same as the optimal threshold for $\text{NEC}_p$ (NB is maximized when $\text{NEC}_p$ is minimized). The threshold for $\text{NEC}_p$ is given by Equation (28) in Appendix A.2, with $c_{21} = 1$, and $c_{12} = p/(1-p)$ resulting in $t = p$, which explains the fact that this parameter is called threshold probability.

### 3.4 LR+

The likelihood ratio for positive results, usually called LR+, is commonly used in diagnostic testing (Pauker & Kassirer, 1975; Šimundić, 2009). As for $F_\beta$ and NB, it is used for non-symmetric binary classification

problems where one of the classes is the class of interest. It is given by:

$$\text{LR+} \quad = \quad \frac{\text{sensitivity}}{1 - \text{specificity}} \tag{22}$$

where, if we assume class $H_2$ is the class of interest, sensitivity $= R_{22}$ and specificity $= R_{11}$. LR+ is always positive and the larger its value, the better the system.

We can write LR+ as a function of a NEC value using the fact that $R_{22} + R_{21} = 1$, and $R_{11} + R_{12} = 1$:

$$\text{LR+} \quad = \quad \frac{1 - R_{21}}{R_{12}} \quad = \quad \frac{1 - \text{NEC}_{\text{b}}}{R_{12}} + 1 \tag{23}$$

where $\text{NEC}_{\text{b}} = R_{12} + R_{21}$ as defined in Section 3.2. So, given two systems with the same $\text{NEC}_{\text{b}}$, the LR+ will favor the one with lower $R_{12}$. Just like for $F_{\beta}$ and MCC, the optimal threshold for LR+ and, hence, the effective costs for which the metric optimizes, will depend on the system's characteristics (this time, through $R_{12}$). We see no reason to prefer this metric over the EC where, if favoring lower values of $R_{12}$ is desired, one can directly and explicitly set $c_{12} > c_{21}$ to the actual costs needed for the application.

### 3.5 Specificity at a target sensitivity

Another approach for measuring performance of binary classification systems, most commonly used in medical diagnosis tasks, is to fix the sensitivity (or recall, or, in our terminology, $R_{22}$) to some pre-defined value and report the specificity ($R_{11}$) corresponding to the threshold that results in that value of sensitivity (Bickelhaupt et al., 2018; Chan et al., 2002; Goldbaum et al., 2002; Qin et al., 2019). Sometimes the opposite is done, fixing the specificity and reporting sensitivity (Souza et al., 2010). These metrics are used when the application of interest imposes a certain target sensitivity or specificity below which the system would be unacceptable. It compares systems by forcing them to operate at that exact level of the target metric and comparing the other metric at that threshold. For concreteness, here we will discuss the first approach where the sensitivity is fixed at a target value and the specificity is reported, calling it SP@SE, though the discussion directly applies to the alternative metric.

SP@SE, while intuitive and easy to explain, has a number of problems. First, consider the scenario where a development dataset is used to compute SP@SE for a fixed sensitivity of 0.95, which corresponds to a certain threshold $t_{\text{dev}}$. Now, when this threshold is used on a new dataset, the sensitivity will most likely no longer be exactly 0.95. What is the value of SP@SE for this new dataset? It is not the specificity at the $t_{\text{dev}}$ threshold because the sensitivity at that threshold is not the target value. We could, instead, set it to be the specificity for a new threshold $t'_{\text{dev}}$ for which the sensitivity is 0.95 in the new dataset. The problem with this latter approach is that it does not address the practical scenario in which the threshold is determined during development and used without change to make decisions on new data.

A second problem with this metric is that, just like $F_{\beta}$ and MCC, it implicitly considers a different set of costs for every system under evaluation since the decision threshold depends on the system. Finally, a third issue with this metric is that it forces all systems to operate at the minimum acceptable sensitivity. When comparing various systems with each other, better systems could potentially operate at better sensitivity values and still achieve good specificity. Yet, this metric does not allow us to adapt the threshold to obtain a better trade off between the two types of error when possible.

In some applications, though, it is important to impose a minimum value of sensitivity. The approach we recommend for these cases is to use a NEC as defined in Equation (4), where the $\alpha$ is determined using some baseline system to achieve (approximately) the desired target sensitivity at the optimal threshold for the corresponding EC. In this way, we translate the original requirement on sensitivity to a metric that can be used to compare systems with each other, select thresholds, and evaluate performance on different datasets without the problems described above related to SP@SE. As an added benefit, this approach makes explicit the implicit assumption on the costs that come with the selected target sensitivity. To find what the ratio of costs corresponding to the selected $\alpha$ is, we simply observe that $c_{12}/c_{21} = \alpha P_2/P_1$, where the priors are those corresponding to the development set. We can then analyze whether the ratio that resulted from setting the sensitivity to the selected target value is, indeed, reasonable for the task. Say, for example, that we are

setting the sensitivity to 0.95 because we believe it is very important to never miss a sample of class $H_2$. Now, after finding the corresponding $\alpha$ we observe that the ratio of costs for which that sensitivity is achieved for our data and system is $c_{12}/c_{21} = 2$. That is, for the desired operating point with sensitivity equal to 0.95 we are, implicitly, assigning more weight to the errors on class $H_1$ than to those of class $H_2$. Upon seeing this, we might decide the selected target sensitivity was, in fact, not a good choice.

### 3.6 Area under the curve and equal error rate

Two very common metrics used to evaluate binary classification systems are the area under the ROC curve (AUC) (Bradley, 1997) and the equal error rate (EER) (Brümmer et al., 2021). These metrics, unlike all other metrics covered in this paper which evaluate categorical decisions, evaluate the quality of the scores. The assumption made by these metrics is that the decisions will be made by thresholding, but no commitment to a specific threshold is made. Under this assumption, two types of curves can be created by sweeping a threshold, making categorical decisions for each case, and computing the resulting $R_{ij}$ values:

- Receiver Operating Characteristic (ROC) curves, which correspond to $R_{22}$ versus $R_{12}$

- Recall vs Precision (PR) curve, which correspond to $R_{22}$ vs $N_{22}/N_{*2}$.

For each of these two curves, the area under the curve (AUC) can be obtained as a summary metric. Further, from the ROC we can find the threshold for which $R_{12}$ equals $R_{21} = 1 - R_{22}$. The value of $R_{12}$ at this threshold is called Equal Error Rate (EER). Interestingly, there is a very close tie between the EER and the total error rate (EC for the 0-1 cost matrix) computed over Bayes decisions: this EC is upper bounded by $\min(\text{EER}, P_1, P_2)$ (Brümmer et al., 2021), as long as calibration is perfect. If calibration is not perfect, this EC can grow without bound, while EER will not. Note that the EER has the same problem as SP@SE: the threshold used to compute it needs to be obtained on the evaluation data itself, resulting in a metric that does not reflect the final performance of a system for which the threshold is determined during the development process.

The ROC and PR curves and the resulting AUCs are invariant to monotonic transformations of the scores since every threshold in the original space corresponds to a new threshold in the transformed space with the same $R_{ij}$ values. As a consequence, these metrics do not assess the effect of threshold selection in the final performance of the system, which could be seen as implicitly assuming that the optimal threshold would always be selected. Further, the AUCs do not reflect the performance that a classification system used to make categorical decisions will have in practice since such systems need to commit to one specific threshold instead of integrating over all possible thresholds.

Hence, while AUC may be a useful metric during the early stages of development, at some point a developer working on a system that will be used to make categorical decisions for a certain application should commit to a specific set of costs that reflect the needs of that application. If the application for which the system will be used is unknown during development or the costs cannot be determined ahead of deployment, then the system needs to be optimized for any possible threshold. In this case, using AUC as a metric would be appropriate, assuming that the threshold for each application can be effectively determined empirically using some development data. Alternatively, in that scenario, performance can be assessed using strictly proper scoring rules (SPSRs) which evaluate the quality of the scores as posterior probabilities (Gneiting & Raftery, 2007; Ferrer & Ramos, 2025). A system developed to have low expected SPSR can be thresholded with the Bayes threshold without the need to tune the threshold for each new set of costs.

Finally, note that the AUC and EER metrics cannot be naturally generalized to the multi-class case, or to binary cases with more than two decisions. Hence, when the task has more than two classes or decisions, these metrics are not applicable.

### 3.7 Multi-class metrics

All the metrics described in this section until this point are originally designed for binary classification. For the multi-class case, the most common metrics in the machine learning literature are accuracy and balanced

accuracy which, as we saw in Section 2.3, are one minus special cases of the EC. Other metrics often used for the multi-class case rely on mapping the original multi-class classification problem into a collection of binary classification problems – a task called multi-label classification (Tsoumakas & Katakis, 2007) – where each binary problem consists of detecting one of the classes. By turning the multi-class problem into a set of binary classification problems, an essential restriction of the multi-class classification task is ignored: the fact that exactly one label should be assigned per sample instead of potentially zero or multiple labels as for the multi-label classification task. This mapping, though, enables the use of metrics given by summaries of binary classification metrics. The macro $F_\beta$ is computed as the average of the $F_\beta$ value over each of the binary classifiers. The $F_\beta$ of macro averages of precision and recall (sometimes also called macro $F_\beta$, producing a good deal of confusion as discussed by Opitz & Burst (2019)) is computed using Equation (12) with the precision and recall calculated as averages over the per-class precision and recall for each of the binary detectors (Grandini et al., 2020). These two metrics inherit all the problems of the binary $F_\beta$ discussed in Section 3.1. Finally, the $F_\beta$ of micro averages of precision and recall is also given by Equation (12), but with the precision and recall computed using the total number of true positives ($N_{22}$), false positives ($N_{12}$) and false negatives ($N_{21}$) across all classes (Zhang & Zhou, 2013). This version of the $F_\beta$, when computed on the output of a multi-class classification system where only one of the binary classifiers can be active for each sample, turns out to be identical to the accuracy (Grandini et al., 2020).

One of the advantages of the EC is that it directly and naturally addresses the multi-class classification task. While the simplest way to define an EC for the multi-class case is to set all the costs to 1, as in the total error rate, or to 1 over the priors, as in the balanced error rate, many problems benefit from the more general definition of the cost matrix. For example, in some scenarios one class may be more important than the others, requiring higher costs. Further, an abstain decision may be included with its corresponding set of costs for each true class. Importantly, as explained in Section 2.4, the EC comes with a principled decision theory for making optimal decisions based on classifiers that output posterior probabilities. This is particularly useful for the multi-class case (or when the number of decisions is larger than two) where threshold tuning is not an option. We illustrate the use of the EC for the multi-class case in Section 4.1.2.

## 4 Empirical comparison

In this section we illustrate and further discuss the concepts explained in the previous sections using empirical results, focusing on comparing the EC with the most widely used metrics in the literature. The first section shows results on various synthetic datasets which allow us to create an arbitrarily large number of samples so that conclusions are not affected by noisy results, and to manipulate the system performance to highlight the different points we wish to discuss. The second section shows results on various real datasets for speech, image, and natural language processing tasks, illustrating how the wrong choice of metric can negatively affect development decisions and highlighting the limitations of the standard metrics.

### 4.1 Results on synthetic datasets

The procedure for generating the synthetic scores is described in detail in Appendix B. Briefly, 100,000 samples are generated with an imbalanced prior distribution, with $P_1 = 0.8$ and equal prior for the rest of the classes. Samples for each class are drawn from a Gaussian distribution where the mean are separated by 1.0 and the variance is 0.2, unless otherwise indicated. Then, perfectly calibrated posteriors are obtained using the known distribution and priors for each class.

### 4.1.1 Comparison of metrics for a two-class dataset

Figure 1 shows results for the 2-class dataset for $F_1$, MCC, and three different NEC with $c_{12} = 1$ and $c_{21} = c$ set to 1, 4, and 16. Since $P_1 = 0.8$, the NEC with $c = 4$ corresponds to the normalized balanced error rate, $\text{NEC}_b = R_{12} + R_{21}$ (this can be verified by plugging these costs and priors in Equation (4)). The left plot in the figure shows the distribution of log posterior odds, $\log(P(H_2|x)/P(H_1|x))$, for each of the two classes. The rest of the plots show the behaviour of different metrics as a function of the decision threshold. The decisions are made by comparing the log odds score for each sample with the corresponding threshold, labelling the sample as class $H_2$ if the score is larger than the threshold and as class $H_1$ otherwise. For each

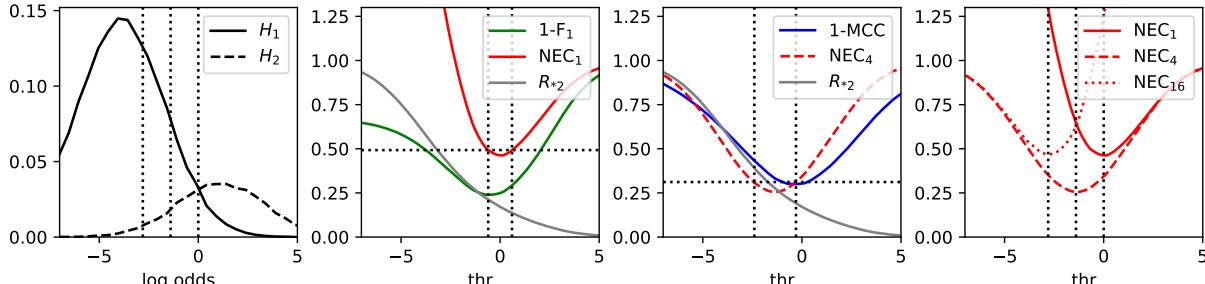

Figure 1: Comparison of metrics for a synthetic dataset with Gaussian per-class distributions and $P_1 = 0.8$. The subindex in the NEC values indicates the value of $c_{21}$, while $c_{12}$ is 1.0 in all cases. The first plot shows the distribution of the log odds for each class scaled by the prior probability of the class. The other three plots show various metrics as a function of the log odds decision threshold. The dashed lines are included only to aid visualization of the various points highlighted in the discussion.

threshold we can then obtain the $N_{ij}$ counts needed to compute all metrics. The dashed lines in these plots correspond to specific decision thresholds discussed below and are included to ease visualization of the metric values for those thresholds.

The second plot in Figure 1 shows a comparison between one minus $F_1$ and $NEC_1$, which are related by Equation (16). The fraction of samples labelled as class $H_2$ for each threshold, $R_{*2}$ is also shown. The figure highlights two thresholds: the one that minimizes $1 - F_1$ and another one that results in the same value of $NEC_1$. As discussed in Section 3.1, between two systems with the same $NEC_1$, the $F_1$ prefers the one with a larger $R_{*2}$, effectively giving more cost to the errors in class $H_2$ by favoring systems that produce more detections of this class. Concretely, given the threshold $t_o$ for which $F_1$ is minimized in a certain dataset, an equivalent NEC for which the Bayes threshold for the log odds is $t_o$ can be defined by using Equation (29) in Appendix A.2, setting $c_{12} = 1$ and $c_{21} = e^{-t_o}$. For the example in Figure 1, the NEC that would lead to the same optimal threshold as the $F_1$ ($t_o = -0.60$) is the one with $c_{21} = 1.8$. That is, in this case, optimizing $F_1$ is equivalent to optimizing NEC assuming that errors in samples of class $H_2$ are 1.8 times more costly than those in samples of class $H_1$. The problem is that this cost is not explicitly selected by the user of the metric and, instead, depends on the distribution of the scores under evaluation.

Figure 2 further illustrates the dependency on the scores of the effective costs corresponding to $F_1$. The figure shows the $F_1$ and two NEC for two different sets of scores: (1) the same scores used for Figure 1, and (2) a new set of scores obtained using the same procedure as for the first but changing the variance for the features from 0.20 to 0.06, simulating a better system. We can see that the $F_1$ is optimized at a different threshold for both datasets, corresponding to an equivalent $c_{21}$ of 1.8 for the original dataset, as discussed above, and an equivalent cost of 1.0 for the easier dataset (with $c_{12}$ always fixed at 1.0). Hence, the $F_1$ effectively optimizes for a different trade-off between error types depending on the system under evaluation. On the other hand, since both datasets are well-calibrated, the Bayes threshold for each of the NECs (Appendix A.2) optimizes the corresponding NEC. This, we believe, is one of the strongest arguments to prefer the EC over the $F_\beta$ (the analysis above was done for $F_1$ but holds for any $F_\beta$). For EC the costs can be explicitly selected and the optimal threshold for calibrated posteriors can be determined theoretically as a function of the costs. On the other hand, for $F_\beta$, the costs are implicit and depend on factors that are outside of our control making the optimal threshold and the effective costs vary across systems.

The third plot in Figure 1 shows one minus MCC and the NEC value with $c = 4$, along with $R_{*2}$. The MCC and this NEC are related by Equation (18). The figure highlights two thresholds: the one that optimizes MCC and a second one that results in the same value of $NEC_4$ as the first one. Comparing the values of MCC across these two threshold, we can observe the behavior explained in Section 3.2: among two thresholds that are equivalent in terms of $NEC_4$, the MCC prefers the one that results in a more imbalanced $R_{*2}$. As for $F_\beta$, the optimal threshold for MCC is determined, in part, by the value of $R_{*2}$. As a consequence, also as

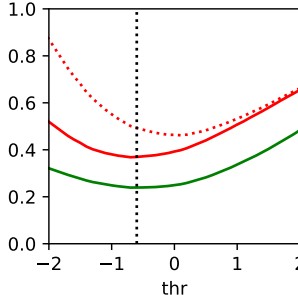 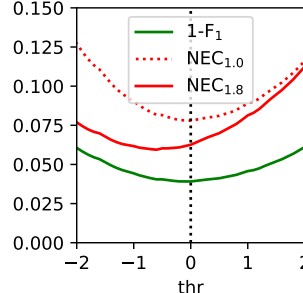

Figure 2: $F_1$ and two NEC metrics with $c_{12} = 1$ and $c_{21}$ indicated as a subindex, for two different datasets. In the left plot, the metrics are computed for the same dataset as in Figure 1. In the right plot they are computed on an easier dataset (note the change in the limits in the y axis). The two NEC are selected to use the effective cost corresponding to $F_1$ for each of the two datasets. For the easier dataset (right), the best threshold for $F_1$ coincides with the Bayes threshold for $NEC_{1.0}$. For the harder dataset (left), the best threshold for $F_1$ coincides with the Bayes threshold for $NEC_{1.8}$.

for $F_\beta$, the equivalent cost implicit in the MCC depends on the system under evaluation giving us no explicit control over the trade-off between types of errors.

Finally, the fourth plot in Figure 1 compares the two NECs from the middle plots with each other and with a third NEC with a larger $c_{21}$. The Bayes threshold for $NEC_c$ is given by $\log(1/c)$ (see Equation (29) in the Appendix): larger values of $c_{21}$ move the threshold to the left decreasing the error rate for class $H_2$ and increasing it for class $H_1$. Note that, as mentioned in Section 2.4, the Bayes threshold is only optimal if the scores are well-calibrated. If this is not the case, the threshold can be tuned empirically, as usually done for any other metric. Alternatively, a calibration stage can be added to the system enabling the use of Bayes thresholds (Filho et al., 2023; Ferrer & Ramos, 2025). In any case, the EC allows us to explicitly select the relative cost between the different error types which directly and transparently affects the resulting metric values and, when the scores are calibrated, the corresponding optimal threshold.

### 4.1.2 Comparison of ECs for a 10-class dataset

In this section we illustrate the use of the EC for a 10-class problem. We compare EC and NEC obtained with the following cost matrices: 1) the 0-1 matrix (**C01**) which corresponds to the error rate, 2) a matrix with $c_{ij} = 1/(KP_i)$ for $i \neq j$ (**CinvP**) which corresponds to the balanced error rate, 3) the C01 matrix with the last row multiplied by 100, representing a use case where errors in the last class are much more costly than errors in other classes (**Cimb**), 4) the C01 matrix with an additional column corresponding to an abstention decision, with cost of 0.05 for all classes (**Cabs1**), and 5) as Cabs1 but with an abstention cost of 0.3 for all classes (**Cabs2**). Table 1 shows the EC and NEC results for these cost matrices for three methods for making categorical decisions: 1) choosing always the best naive decision for each cost matrix, 2) making "argmax" decisions, i.e., choosing the class with the largest score for each sample, regardless of the cost matrix (for this row, decisions are the same for all cost matrices), and 3) making the Bayes decisions corresponding to each cost matrix using Equation (7).

Comparing the last two lines in this table we see that, as theory shows, Bayes decisions result in better performance than argmax decisions, except for the C01 matrix for which Bayes decisions are the argmax decisions and, hence, those two results are the same. Further, the table highlights the benefit of normalization. While the EC might be misleadingly low for some cost matrices due to the imbalanced priors and the selected cost values, the NEC solves this problem by using the EC of the best naive system as baseline with which to normalize, resulting in a more interpretable metric.

The best naive decision depends on the cost matrix as determined by Equation (3). For C01, the class with the largest prior, $H_1$, is the least-costly naive decision. For CinvP, all decisions are equally good due to the way the costs are defined. For Cimb, the least costly decision is $H_{10}$ due to the large cost assigned to errors in that class. Finally, the least-costly decision for Cabs1 is to abstain, while for Cabs2 it is $H_1$ (see Appendix A.3

Table 1: EC and NEC for various costs matrices defined in Section 4.1.2 for systems that make the best naive decision (the decision made by this system is indicated in parenthesis under the cost matrix name), argmax decisions, and Bayes decisions. The accuracy and balanced accuracy can be obtained as one minus the EC for C01 and CinvP, respectively. For the two cost matrices with an abstain decision, the percentage of abstain decisions made by the system is also listed (abs).

| Naive Decision → Decision algorithm | C01 (class $H_1$) | | CinvP (any class) | | Cimb (class $H_{10}$) | | Cabs1 (abstain) | | | Cabs2 (class $H_1$) | | |
|---|---|---|---|---|---|---|---|---|---|---|---|---|
| | EC | NEC | EC | NEC | EC | NEC | EC | NEC | abs | EC | NEC | abs |
| Naive | 0.20 | 1.00 | 0.90 | 1.00 | 0.98 | 1.00 | 0.05 | 1.00 | 100 | 0.20 | 1.00 | 0 |
| Argmax | 0.06 | 0.32 | 0.28 | 0.31 | 0.36 | 0.37 | 0.06 | 1.29 | 0 | 0.06 | 0.32 | 0 |
| Bayes | 0.06 | 0.32 | 0.23 | 0.26 | 0.08 | 0.08 | 0.02 | 0.35 | 25 | 0.06 | 0.28 | 7 |

for the expression for Bayes decisions and for the best naive decisions for the Cabs cost matrices). As shown in the column "abs" for these last two cost matrices, a larger percentage of abstain decisions are made with the Bayes decision algorithm for Cabs1 than Cabs2, due to the fact that Cabs1 assigns a lower abstention cost than Cabs2. Finally, we can see that while the lower abstention cost in Cabs1 results in a lower EC value than for Cabs2, the NEC shows the opposite trend. This happens because the best naive system is different for both ECs, resulting in different baselines used for normalization.

Note that the scores used for these synthetic experiments are, by design, perfectly calibrated, so Bayes decisions are optimal in this dataset. In practice, this might not be the case for a certain dataset of interest. Unlike in the scenario of the previous section where we could simply tune a threshold instead of using Bayes decision theory, for the multi-class case or, more generally, when the number of decisions is larger than 2, tuning the decision regions is not trivial (though attempts have been made in this direction, e.g., by O'Brien et al. (2008)). In those scenarios, we recommend making sure that the system outputs are well-calibrated (Filho et al., 2023; Ferrer & Ramos, 2025) and then applying Bayes decision theory as done in this section.

## 4.2 Results on real datasets

In this section, we show results on real datasets for a number of different tasks corresponding to speech, image and natural language processing tasks. The datasets and systems used to generate the scores studied in this section are described in Appendix C. For each system, we show two sets of results obtained on the raw scores as they come out of the system, and on calibrated scores. The calibrated scores are obtained with linear logistic regression using 5-fold cross-validation on the test data. Finally, for some datasets, we show results for resampled versions where some of the classes are downsampled to obtain a dataset with priors different from the original ones to show the effect this change has on the metrics. The details on the calibration process and the priors for each dataset are listed in Appendix C.

Table 2 shows the results on all these datasets for three NECs, accuracy (ACC), two multi-class extensions of the $F_1$, and, for the binary cases, the standard binary $F_1$, AUC, and EER. The multi-class extensions of the $F_1$ are the average of the $F_1$ over all classes (MC1) and the $F_1$ of the average recall and precision (MC2) (see Section 4.1.2). To facilitate comparisons, for those metrics that increase as performance improves (AUC, ACC, and the three $F_1$), we report one minus the metric value. Decisions for the three NEC metrics are made using Bayes decision theory. For all three $F_1$s and ACC, decisions are made with the argmax approach, i.e., by selecting the class with the highest score, as is usually done in the literature. For AUC and EER decisions are not made since those metrics evaluate scores rather than categorical decisions.

Comparing the results on raw and calibrated scores we can see that AUC and EER are insensitive to the linear logistic calibration applied in these experiments since, as explained in Section 3.6, these metrics are invariant to monotonic transformations of the scores. The small variation in AUC and EER results observed in some cases after calibration is due to the cross-validation approach, which results in different transformations being applied to different subsets of the data. We can also see that, as explained in Section 3.6, when the scores are

Table 2: Various metrics on real scores for different speech, image and natural language processing datasets. We report three NEC values, EER, 1-ACC, 1-$F_1$ for three version of $F_1$, and 1-AUC (lower values are better for all cases). Note that 1-ACC corresponds to the EC for C01 (EC values are not shown in the table to reduce clutter). The term 'ResN' in some of the dataset names identifies a resampled version of a dataset (see Table 3). The colored entries highlight comparisons discussed in the text.

| Dataset | System | proc | NEC C01 | CinvP | Cimb | 1-ACC | 1-$F_1$ MC1 | MC2 | Bin | 1-AUC | EER |
|---|---|---|---|---|---|---|---|---|---|---|---|
| SST2 | GPT2-4sh | raw | 0.996 | 0.992 | 1.000 | 0.497 | 0.667 | 0.667 | 0.332 | 0.048 | 0.116 |
| | | cal | 0.226 | 0.225 | 0.921 | 0.113 | 0.486 | 0.329 | 0.115 | 0.048 | 0.116 |
| | GPT2-0sh | raw | 0.828 | 0.826 | 1.000 | 0.414 | 0.667 | 0.667 | 0.294 | 0.072 | 0.152 |
| | | cal | 0.310 | 0.308 | 0.907 | 0.155 | 0.571 | 0.363 | 0.156 | 0.072 | 0.152 |
| SST2-Res1 | GPT2-0sh | raw | 1.931 | 0.991 | 1.000 | 0.579 | 0.769 | 0.769 | 0.493 | 0.074 | 0.159 |
| | | cal | 0.451 | 0.306 | 1.150 | 0.135 | 0.595 | 0.379 | 0.238 | 0.074 | 0.158 |
| SST2-Res2 | GPT2-0sh | raw | 3.298 | 1.000 | 1.000 | 0.660 | 0.833 | 0.833 | 0.625 | 0.077 | 0.150 |
| | | cal | 0.526 | 0.305 | 1.071 | 0.105 | 0.597 | 0.380 | 0.294 | 0.079 | 0.155 |
| SITW | XvPLDA | raw | 0.324 | 0.071 | 0.095 | 0.002 | 0.415 | 0.295 | 0.167 | 0.002 | 0.021 |
| | | cal | 0.306 | 0.043 | 0.062 | 0.002 | 0.302 | 0.236 | 0.166 | 0.002 | 0.022 |
| SITW-Res1 | XvPLDA | raw | 0.195 | 0.070 | 0.181 | 0.019 | 0.098 | 0.090 | 0.107 | 0.002 | 0.022 |
| | | cal | 0.117 | 0.044 | 0.126 | 0.012 | 0.154 | 0.134 | 0.059 | 0.002 | 0.022 |
| SITW-Res2 | XvPLDA | raw | 0.192 | 0.103 | 0.352 | 0.038 | 0.056 | 0.054 | 0.106 | 0.002 | 0.022 |
| | | cal | 0.083 | 0.045 | 0.166 | 0.017 | 0.121 | 0.108 | 0.042 | 0.002 | 0.022 |
| FVCAUS | XvPLDA | raw | 3.915 | 0.315 | 0.362 | 0.067 | 0.566 | 0.362 | 0.662 | 0.000 | 0.001 |
| | | cal | 0.012 | 0.003 | 0.005 | 0.000 | 0.017 | 0.016 | 0.006 | 0.000 | 0.001 |
| CIFAR-1vsO | Resnet20 | raw | 0.420 | 0.055 | 0.055 | 0.004 | 0.223 | 0.187 | 0.223 | 0.003 | 0.022 |
| | | cal | 0.430 | 0.041 | 0.041 | 0.004 | 0.312 | 0.239 | 0.230 | 0.003 | 0.023 |
| CIFAR-2vsO | Resnet20 | raw | 0.700 | 0.150 | 0.151 | 0.007 | 0.327 | 0.262 | 0.368 | 0.013 | 0.056 |
| | | cal | 0.560 | 0.108 | 0.109 | 0.006 | 0.393 | 0.289 | 0.346 | 0.013 | 0.056 |
| IEMOCAP | W2V2 | raw | 0.504 | 0.434 | 0.839 | 0.349 | 0.567 | 0.445 | - | - | - |
| | | cal | 0.494 | 0.428 | 0.804 | 0.342 | 0.661 | 0.489 | - | - | - |
| IEMOCAP-Res1 | W2V2 | raw | 0.498 | 0.457 | 0.665 | 0.340 | 0.632 | 0.460 | - | - | - |
| | | cal | 0.471 | 0.435 | 0.620 | 0.322 | 0.598 | 0.442 | - | - | - |
| AGNEWS | GPT2-0sh | raw | 0.780 | 0.780 | 1.015 | 0.585 | 0.897 | 0.694 | - | - | - |
| | | cal | 0.378 | 0.378 | 1.179 | 0.283 | 0.769 | 0.567 | - | - | - |
| AGNEWS-Res1 | GPT2-0sh | raw | 0.742 | 0.946 | 0.998 | 0.497 | 0.992 | 0.714 | - | - | - |
| | | cal | 0.277 | 0.365 | 0.584 | 0.186 | 0.423 | 0.336 | - | - | - |
| CIFAR10 | Resnet20 | raw | 0.082 | 0.082 | 0.273 | 0.074 | 0.087 | 0.085 | - | - | - |
| | | cal | 0.083 | 0.083 | 0.184 | 0.075 | 0.126 | 0.116 | - | - | - |
| | Vgg19 | raw | 0.068 | 0.068 | 0.357 | 0.061 | 0.064 | 0.064 | - | - | - |
| | | cal | 0.069 | 0.069 | 0.186 | 0.062 | 0.093 | 0.087 | - | - | - |
| | RepVgg-a2 | raw | 0.053 | 0.053 | 0.169 | 0.047 | 0.053 | 0.052 | - | - | - |
| | | cal | 0.052 | 0.052 | 0.129 | 0.047 | 0.073 | 0.069 | - | - | - |
| CIFAR100 | Resnet20 | raw | 0.315 | 0.315 | 0.357 | 0.312 | 0.319 | 0.314 | - | - | - |
| | | cal | 0.317 | 0.317 | 0.351 | 0.314 | 0.329 | 0.321 | - | - | - |
| | Vgg19 | raw | 0.264 | 0.264 | 0.465 | 0.261 | 0.260 | 0.258 | - | - | - |
| | | cal | 0.266 | 0.266 | 0.358 | 0.263 | 0.271 | 0.260 | - | - | - |
| | RepVgg-a2 | raw | 0.227 | 0.227 | 0.351 | 0.225 | 0.230 | 0.223 | - | - | - |
| | | cal | 0.228 | 0.228 | 0.341 | 0.226 | 0.229 | 0.223 | - | - | - |

calibrated, the EER value is (approximately, since we are dealing with limited-size datasets) an upper bound for the EC value corresponding to the C01 cost (1-ACC) (see brown entries in the table).

Results across versions of the same dataset with different priors (SST2, SITW, IEMOCAP, and AGNEWS) illustrate how the variation in priors can greatly affect the NEC values for C01 and Cimb, as well as $F_1$ and ACC (cyan entries highlight those comparisons for SST2 and AGNews for calibrated scores). On the other hand, AUC and EER are (again, approximately, since we are dealing with limited-size datasets) unaffected by the priors. Further, for the calibrated systems, the NEC for CinvP is also (approximately) unaffected. This is, in fact, exactly was this cost matrix is designed to do. Its costs are determined for each dataset as the inverse of the priors so that the final metric is immune to the priors in the dataset as long as the scores are calibrated and Bayes decisions are made or the threshold is determined by optimizing the metric on a development set. A more general version of this type of cost matrix designed to be independent of the priors of the evaluation set was used by Godau et al. (2023) to address the problem of tasks where the priors in deployment are different from the priors in the development data.

Comparing the three flavors of $F_1$ with each other we can see that they are often quite different (see, for example, the blue entries in Table 2). The difference stems in part from the fact that, while the binary $F_1$ prioritizes one of the classes, the multi-class versions are symmetric across classes. Yet, despite both multi-class versions being symmetric in the classes, they differ from each other and they can even lead to contradicting conclusions, as discussed by Opitz & Burst (2019). Importantly, none of the $F_1$ variants (or, more generally, $F_\beta$ variants) offers direct control on the trade off between error types, like the EC does. Hence, there is no principled way to select between the different variants based on the needs of the task. Further, we can see that the multi-class $F_1$ metrics sometimes degrade after calibration (see SITW-Res1, SITW-Res2, IEMOCAP and some of the CIFAR results – two of these examples are marked in orange in the table). These results illustrate the fact that argmax decisions are not necessarily optimal for the multi-class $F_1$ metrics, even if the scores are perfectly calibrated since, if making argmax decisions with calibrated scores was optimal, no other way of making decisions would lead to better results.[9] The fact that, in some cases, using the raw scores to make argmax decisions leads to better results than using the calibrated scores implies that a better decision strategy exists, and, hence, argmax decisions from calibrated scores are not guaranteed to optimize these metrics. As far as we know, there is no principled way to optimize decisions for these metrics other than empirical optimization. This, though, is very rarely done for the multi-class case given the difficulty involved in optimizing decision regions in more than one dimension. In contrast, if scores are calibrated, for every NEC metric there is a corresponding decision strategy that is guaranteed to be optimal (Ferrer & Ramos, 2025).

Finally, a general observation from Table 2 is that systems can be ranked differently depending on the metric used to rank them. For example, while calibration improves the Cimb NEC value for CIFAR10 Resnet20 system, it does not affect the other NECs or the ACC, and it actually degrades the multi-class $F_1$ metrics (see red entries in the table). This also holds for the other two systems on CIFAR10. Also, comparing results of the Resnet20 and Vgg19 calibrated scores for the CIFAR10 and CIFAR100 datasets we can see that Vgg19 is better than Resnet for both datasets in terms of all metrics except Cimb NEC, for which the Resnet20 system is slightly better (see green entries and corresponding rows for CIFAR10). If Cimb NEC was the metric of interest for this task (i.e., if errors in class $H_{10}$ costed 100 times more than errors on other classes), then assessing the system performance based on ACC or one of the $F_1$ could lead to suboptimal development decisions. For example, in this case, using ACC would lead us to conclude that calibration is not necessary and that the Vgg19 system is better than the Resnet20 system. In contrast, for Cimb NEC, the metric that reflects the needs of the task, calibration gives a gain and Resnet20 is slightly better than Vgg19, having the additional advantage of being 100 times smaller (see Appendix C). These are just some examples that highlight the importance of selecting an appropriate metric for the scenario of interest, a task that is greatly facilitated by the EC metric.

---

[9]This is also true for the binary $F_1$ metric, as can be seen in Figure 1 where the optimal threshold for $F_1$ is not the one that corresponds to argmax decisions (which is zero in the log-odds domain).

## 5 Conclusions

In this work we focus on the problem of evaluating the quality of categorical decisions generated by a machine learning classifier. We argue that the expected cost (EC), a classic evaluation function proposed decades ago, should be the primary metric to assess the performance of categorical decisions for a certain application of interest. The EC is based on the simple assumption that each decision made by the system can be assigned a cost which depends on the true class of the sample. The expectation of the cost over the data can then be used as a metric to assess the performance of a system. The EC can be customized to any application for which categorical decisions are required by adapting the costs, and optionally the class priors, according to the needs of the application.

While two special cases of the EC, the error rate and the balanced error rate are widely used in the machine learning literature, its general form is only rarely leveraged. Instead, metrics like the $F_\beta$ score or the MCC are used for applications in which the error rate is deemed inadequate for performance assessment. This is often the case in applications with large class imbalance and where errors in the minority class are considered more important than those in the majority class. In this work, we argue that the EC is a better metric than those ad-hoc alternatives. The advantages of the EC can be summarized as follows:

- It is defined based on the simple and general assumption that each combination of true class and decision can be assigned a corresponding fixed cost. In contrast, for other classification metrics, the trade-off between different types of error cannot be directly controlled.

- The class prior distribution can be considered a parameter of the metric. This is useful in cases in which the priors in the evaluation data do not coincide with those expected at deployment.

- The same general EC expression can be used to evaluate both binary and multi-class classification systems in a principled way.

- Unlike other classification metrics, the EC does not assume that the set of possible decisions is the set of classes. Instead, the decisions can be the actions that may be taken by the user of the system.

- When normalized by the EC of the best naive systems for the selected costs and priors, the resulting metric is easily interpretable as a relative value with respect to the performance of the best naive system.

- Optimization of categorical decisions can be done using Bayes decision theory as long as the system scores are well-calibrated, eliminating the need for ad-hoc solutions. Alternatively, though, when calibrated scores are not available, the decision strategy can be based on threshold or region tuning, as for any other classification metric.

The need to choose costs may seem like a drawback of the EC compared to other metrics like the F1 score or the MCC which have no parameters to select. Yet, is it important to note that, while these alternative metrics do not explicitly require the selection of costs, they effectively impose a trade off between different types of error. This is easy to see for the binary case, where decisions are made by thresholding, by noting that the threshold that optimizes a metric for a certain set of scores is the Bayes threshold for some EC. Hence, optimizing the original metric is equivalent to optimizing the related NEC. Effective costs are implicitly considered in the optimization. Yet, those costs are selected without our control and depend on the system under evaluation. The EC, on the other hand, allows for an explicit choice of costs. Once the costs are chosen, the trade-off between different types of errors is fixed and transparent and does not depend on the system under evaluation. While the costs may be hard to specify in some cases, we argue that having control over them is better than letting the metric implicitly choose them.

In conclusion, we argue that the EC, or its normalized version, should be the preferred metric for evaluating categorical decisions for a given application of interest. This work is accompanied by a python library of methods to compute the EC, along with utility methods for making Bayes decisions and calibrating scores. The code includes notebooks that produce the results in this paper. We hope this work will facilitate the wider adoption by the machine learning community of this classic, simple, and well-motivated metric.

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

## A  Special cases of Bayes decisions theory

In this section we discuss three special cases of Bayes decision theory, the argmax rule which is obtained when the cost function is given by the 0-1 cost matrix, the thresholding rule obtained for the binary case with only two decisions, and the case in which the cost function includes an abstention decision.

### A.1  The 0-1 cost matrix

A special scenario of Bayes decision theory applied to the EC arises when the cost matrix is the 0-1 square matrix given by one minus the identity matrix. In this case, the EC reduces to the error rate or one minus the accuracy (see Section 2.3). The Bayes decisions for this EC can be obtained by replacing $c_{ij} = 0$ for $i = j$ and $c_{ij} = 1$ for $i \neq j$ in Equation (7) in Section 2.4, which gives:

$$d_B(x) \quad = \quad \underset{\hat{\jmath}}{\arg\min} \sum_{i=1|i\neq\hat{\jmath}}^{K} P(i|x) \tag{24}$$

$$= \quad \underset{\hat{\jmath}}{\arg\min} \, 1 - P(\hat{\jmath}|x) \tag{25}$$

$$= \quad \underset{\hat{\jmath}}{\arg\max} \, P(\hat{\jmath}|x). \tag{26}$$

This is the standard argmax decision rule used in most of the literature in machine learning for making decisions in a multi-class setting. We can now see that this rule is optimal for the 0-1 cost matrix. For any other cost matrix, the general expression in Equation (7) should be used to make decisions.

### A.2  Binary classification with two decisions

Another special case of Bayes decision theory corresponds to binary classification when the cost matrix is square (i.e., when there is only two possible decisions) with zeros in the diagonal. This is the simplest and

most common case in which there is one correct zero-cost decision for each class. In that case, Bayes decisions reduce to:

$$d_B(x) = \begin{cases} H_2, & \text{if } c_{12}P(H_1|x) < c_{21}P(H_2|x) \\ H_1, & \text{otherwise} \end{cases} \tag{27}$$

Using the fact that $P(H_2|x) = 1 - P(H_1|x)$, this rule becomes:

$$d_B(x) = \begin{cases} H_2, & \text{if } P(H_2|x) > t \\ H_1, & \text{otherwise} \end{cases} \quad \text{where} \quad t = c_{12}/(c_{12} + c_{21}) = 1/(1 + c_{21}/c_{12}) \tag{28}$$

Sometimes we prefer to work with posterior log odds because the logarithm mitigates numerical precision problems and because their distributions are nicer for plotting (see an example in Figure 1 of the main text). When using log odds as scores, the decision rule becomes:

$$d_B(x) = \begin{cases} H_2, & \text{if } \log(P(H_2|x)/P(H_1|x)) > t_{\text{logodds}} \\ H_1, & \text{otherwise} \end{cases} \quad \text{where} \quad t_{\text{logodds}} = \log(c_{12}/c_{21}) \tag{29}$$

In summary, for binary classification with one zero-cost decision per class, Bayes decisions simply consist of comparing the posterior for one of the classes, or some function of the posteriors, like the logodds, with a specific threshold which is a function of the ratio between the two costs. While to optimize metrics like the $F_\beta$ or MCC we need to sweep a threshold and choose the best one for that metric, for the EC we simply need to plug the costs into Equation (28) or (29) to get the optimal threshold.

Note that, as discussed in the main text, Bayes decisions are optimal as long as the classifier is calibrated for the selected cost. If this is not the case, for the binary case above, the threshold can be tuned as for any other metric. For the general case, including multi-class cases or binary cases with more than two decisions (non-square cost matrix), the posteriors can be calibrated to enable the use of Bayes decision theory (Filho et al., 2023). For an in-depth discussion of Bayes decision theory and calibration, please refer to (Ferrer & Ramos, 2025).

### A.3 Cost matrices with an abstention option

Finally, we derive the expression for the Bayes decisions when the set of decisions is given by $\{H_1, \ldots, H_K, \text{abstain}\}$ (the list of classes plus an abstain decision) and the cost matrix has the following form:

$$\mathbf{C} = [1 - I \ \ \alpha\mathbf{1}] \tag{30}$$

where $I$ is the identity matrix, so that $1 - I$ is the 0-1 cost matrix with a cost of 0 in the diagonal and a cost of 1 elsewhere, and where $\alpha\mathbf{1}$ is a vector of ones multiplied by a scalar $\alpha$ which determines the cost of abstention, set to be equal for all classes. Replacing these costs in Equation (7) we get the following expression:

$$d_B(x) = \arg\min_{\hat{j}} \sum_{i=1}^{K} c_{i\hat{j}} \, P(H_i|x) = \arg\min_{\hat{j}} \begin{cases} \sum_{i=1|i\neq\hat{j}}^{K} P(H_i|x) = 1 - P(H_{\hat{j}}|x), & \text{if } \hat{j} \neq \text{abstain} \\ \sum_{i=1}^{K} \alpha P(H_i|x) = \alpha, & \text{if } \hat{j} = \text{abstain} \end{cases} \tag{31}$$

Or, equivalently:

$$d_B(x) = \arg\max_{\hat{j}} \begin{cases} P(H_{\hat{j}}|x), & \text{if } \hat{j} \neq \text{abstain} \\ 1 - \alpha, & \text{if } \hat{j} = \text{abstain} \end{cases} \tag{32}$$

Hence, the system will abstain if all the posteriors output by the system are smaller than $1 - \alpha$.

We can now easily derive the expression for the best naive decision which is the Bayes decision when the system outputs always the prior probability of each class in place of the posteriors.[10] Doing that, we obtain:

$$d_{\text{naive}} = \arg\max_{\hat{j}} \begin{cases} P_{\hat{j}}, & \text{if } \hat{j} \neq \text{abstain} \\ 1 - \alpha, & \text{if } \hat{j} = \text{abstain} \end{cases} \tag{33}$$

---

[10]This same procedure can be used to arrive at the general expression for the EC of the best naive system, Equation (3) in Section 2.1, which was derived in a different way in that section.

Hence, the best naive decision is to abstain when all priors are smaller than $1 - \alpha$. For our multiclass experiments in Section 4 where $P_1 = 0.8$ and $P_i = 0.022$ for $i \neq 1$, the best naive decision is to abstain for Cabs1 where $1 - \alpha = 0.95$, but not for Cabs2, where $1 - \alpha = 0.7$. In that case, the best decision is $H_1$, which has a prior of 0.8.

## B  Synthetic datasets for experiments

Here we describe the procedure used to create synthetic datasets for the experiments in this paper. Given a number of classes $K$, a prior for class $H_1$ of $P_1$, and a per-class variance $\sigma$, we proceeded as follows:

1. Set the class priors to $P_1$ for class $H_1$, and $P_i = (1 - P_1)/(K - 1)$ for classes 2 through $K$. In our experiments, $P_1$ is set to 0.8.

2. Set the total number of samples, $N = 100,000$ and determined the number of samples for each class, $N_i$ as the closest integer to $NP_i$.

3. Generated $N_i$ samples using a unidimensional Gaussian distribution $\mathcal{N}(\mu_i, \sigma)$, with mean $\mu_i = i - 1$ and variance $\sigma$, taking these samples as the input features, $x$.

4. Computed the likelihoods for each class for each generated sample according to the class distributions used to draw these samples, i.e., $p(x|H_i) \sim \mathcal{N}(\mu_i, \sigma)$.

5. Finally, we computed the posteriors as:

$$P(H_i|x) = \frac{p(x|H_i)\ P_i}{p(x)} = \frac{p(x|H_i)\ P_i}{\sum_j p(x|H_j)\ P_j}. \tag{34}$$

where the $p(x|h)$ are the likelihoods computed in step 4 and $P_i$ is the prior for class $H_i$ defined in step 1. With this procedure, the posteriors are perfectly calibrated for the test data.

The value of $\sigma$ was set to 0.20 for all results in this work, except for those in Figure 2 where two datasets are used, one with $\sigma = 0.20$ and an easier one with $\sigma = 0.05$.

## C  Real datasets for experiments

For the experiments with real datasets in Section 4.2, we use a variety of datasets from speech, image, and natural language processing tasks.

SST2 (Socher et al., 2013) is a natural language processing dataset where the task is to decide whether a certain text has positive or negative sentiment. AGNEWS (Gulli, 2005; Zhang et al., 2015) is another natural language processing dataset where the task is to classify news into 4 different classes. The scores for these datasets were produced with the GPT-2 model using the code provided in `https://github.com/LautaroEst/efficient-reestimation` using zero-shot prompts.

SITW (McLaren et al., 2016) and FVCAUS (Morrison et al., 2015; 2012) are two speaker verification datasets where the task is to decide whether two audio samples belong to the same speaker or not. To obtain scores for these two datasets, we ran an X-vector PLDA system using the code provided in `https://github.com/luferrer/DCA-PLDA`.

IEMOCAP is a speech processing dataset where the task is to classify each speech sample into a set of emotions: angry, happy, sad, and neutral. The scores were downloaded from `https://github.com/habla-liaa/ser-with-w2v2/tree/master/experiments/w2v2PT-fusion`.

Finally, CIFAR10 and CIFAR100 are two image processing datasets where the task is to classify the object in an image into one of 10 or 100 classes, respectively. The scores for these datasets were obtained using the code available in `https://github.com/chenyaofo/image-classification-codebase`. We run three

Table 3: Number of classes, number of samples, and class priors for each dataset included in our experiments.

| Dataset | #Classes | #Samples | Priors |
|---|---|---|---|
| SST2 | 2 | 1821 | 0.50 0.50 |
| SST2-Res1 | 2 | 1301 | 0.70 0.30 |
| SST2-Res2 | 2 | 1140 | 0.80 0.20 |
| SITW | 2 | 721788 | 0.99 0.01 |
| SITW-Res1 | 2 | 36580 | 0.90 0.10 |
| SITW-Res2 | 2 | 18290 | 0.80 0.20 |
| FVCAUS | 2 | 114072 | 0.98 0.02 |
| CIFAR-1vsO | 2 | 10000 | 0.99 0.01 |
| CIFAR-2vsO | 2 | 10000 | 0.99 0.01 |
| IEMOCAP | 4 | 5473 | 0.20 0.29 0.31 0.20 |
| IEMOCAP-Res1 | 4 | 3483 | 0.32 0.32 0.32 0.05 |
| AGNEWS | 4 | 7600 | 0.25 0.25 0.25 0.25 |
| AGNEWS-Res1 | 4 | 5757 | 0.33 0.33 0.33 0.01 |
| CIFAR10 | 10 | 10000 | 0.10 for all classes |
| CIFAR100 | 100 | 10000 | 0.01 for all classes |

models for each of the two datasets: resnet20, vgg19, and repvgg_a2. These models have approximately 0.27 million, 20 million, and 27 million parameters, respectively.

From the CIFAR100 Resnet20 scores, we also created binary classification tasks for the detection of one specific class versus all others, using the score provided by the system for that class and 1 minus that score for the "other" class. We call these scores CIFAR-XvsO, where X identifies the target class.

For all sets of scores, we show the results for the raw scores, as they come out of the system, and for scores calibrated with a logistic regression model. This model maps the logarithm of the raw posteriors generated by the systems using the following expression:

$$\hat{s} = \mathrm{softmax}(\alpha \log(s) + \beta) \tag{35}$$

where $s$ is the vector of raw posteriors, $\hat{s}$ is the calibrated vector of posteriors, $\alpha$ is scalar and $\beta$ is a vector. The parameters $\alpha$ and $\beta$ were trained by minimizing the cross-entropy. In this work, the models were trained and the scores generated using a 5-fold cross-validation approach on the test data.

Resampled versions of some of the datasets, identified by a suffix ResN are also used to achieve different priors from the ones in the original dataset. Table 3 shows the priors and total number of samples for all the datasets used in Section 4.2.

