# OpenReview forum: "No Need for Ad-hoc Substitutes: The Expected Cost is a Principled All-purpose Classification Metric"
_TMLR — Accepted by TMLR_

### Review · Reviewer_BNMf · 2024-07-02

**Summary Of Contributions:**

The paper seeks to unify some commonly used metrics for classification (precision, recall, Matthews Correlation Coefficient etc) into a single decision-theoretic framework. Along the way it provides a breif introduction to decision theory for the ML practitioner who uses classification metrics in their table of results without awareness of the foundations, including some worked examples.

**Audience:**

Yes

**Broader Impact Concerns:**

n.a

**Claims And Evidence:**

Yes

**Requested Changes:**

I would recommend that the authors check their work against the Dyrland references above (although since this is a resubmission, those articles may have been released while this article was under submission, so I do not wish to insist to firmly upon this, or the article might get stuck in "revision hell.")

It is possible that I am missing some minor errors or imperfections in this article, but nothing leaps out at me as a major blocker. I will revisit this if I notice anything in my next re-read of the article.

**Strengths And Weaknesses:**

Strengths:

* Putting a bunch of machine-learning metrics in a common decision-theoretic foundation is helpful for many reasons, including

  1. it reveals the hidden cost function in classic metrics
  2. It suggests other metrics that could be constructed using different choices of cost function and prior.
  3. it reveals the class-imbalance "problem" to be self-inflicted

* Actually doing the decision-theoretic analysis of these metrics reveals some surprising features of existing metrics, e.g. $F_{\beta}$ and MCC have no corresponding decision theory (!)

* this is IMO a pretty well-written introduction to decision theory, and in particular the important special case of discrete decisions.

Weaknesses:

The framing seems odd to someone with my background, coming from statistics into ML, where we would assume *all* the metrics have a hidden decision-theoretic basis. Nonetheless, I acknowledge that this might not be part of the ML curriculum, and in fact my own prideful statisticians assumptions seem to have been faulty, since putting some classifiers into a utility-maximizing framework seems to be non-trivial, so putting this in the language of DeGroot or Savage is not so easy.

All that said, much of the paper repackages well-known results. In the ML literature — they mention Bishop 2006 and Hastie et al 2001 — which are part of the ML canon. I am aware also that Dyrland and co-authors have published some work which looks rather similar.

* https://arxiv.org/abs/2302.10578
* https://arxiv.org/abs/2302.12006

---

> ### Author Response · Authors · 2024-07-03
> **Response to review**
>
> Thank you very much for the thoughtful comments. We empathize with your surprise about the fact that the ML community is using metrics that are not grounded in decision theory. Yet, we believe our results (and also those of Dyrland et al.) clearly show this to be the case. This is quite troublesome since, as we discuss in our paper, systems developed based on such metrics may be quite suboptimal in practice. Bringing this problem to the attention of the ML community is the main motivator of our work.
>
> We are very thankful for the pointer to the excellent work by Dyrland et al. We had not found those papers, despite having done a very extensive literature search at the time of the first submission last year. Dyrland's work is, indeed, quite in line with ours, calling the community to stop using ad-hoc metrics in favor of a metric grounded on decision theory. Yet, while the main idea behind that work is consistent with ours, there are several differences that, we believe, make our work complementary to that of Dyrland et al. and, hopefully, valuable to the community:
>
> * While Dyrland et al. show that F1, MCC and other metrics are not special cases of the EC, we go a step further and show *how* the expressions of those metrics are related to the expression of the EC, providing an intuition on exactly how those metrics differ from the EC and why they are suboptimal.
>
> * We discuss the issue of the possible mismatch in priors between the available testing data and the deployment scenario and show how the EC is able to elegantly handle such cases by considering the priors as parameters of the metric. This, in turn, enables us to see the balanced error rate as a special case of the EC. In contrast, Dyrland et al. argue that balanced accuracy (one minus the balanced error rate) is not a special case of the expected utility, since they do not allow for the utilities to depend on the priors. We believe this is too restrictive since decision theory still holds when the priors are taken as parameters.
>
> * We introduce the normalized EC, a more easily interpretable metric than the raw EC which values depend on the scale and center selected for the costs, an often arbitrary choice. A normalized EC of 1.0 implies that the system performs as badly as a naive system that only has access to the priors, providing a useful reference for understanding whether a system is good or bad.
>
> * Finally, Dyrland only compares the metrics for binary problems on synthetic confusion matrices. We show a wider range of empirical results, comparing the metrics as a function of the decision threshold for a binary case and including results on multi-class problems.
>
> We will add references to Dyrland et al. in our paper where appropriate.

---

### Review · Reviewer_V7Xp · 2024-08-16

**Summary Of Contributions:**

This paper evaluates the conventional wisdom of classification metrics. It makes the claim that the expected cost (EC) is superior to other classification metrics, like the F-beta score and MCC. It supports the claim by providing the theoretical and empirical results, including demonstrating how it can make optimal decisions, and be more interpretable.

**Audience:**

Yes

**Broader Impact Concerns:**

There is no impact statement section in the paper and should be added.

**Claims And Evidence:**

Yes

**Requested Changes:**

My main suggestions are around addressing the points listed in the weakness section. In addition to that, it could be beneficial for the paper to answer:

1. Could the paper give some insights into the impact of miscalibration on the EC metric's performance? As presented in the paper, the effectiveness of the EC metric relies heavily on the assumption that the classifier’s output probabilities are perfectly calibrated. While the paper does acknowledge this and suggests post-hoc calibration, it could consider scenarios where calibration is not perfect.
2. The paper emphasizes the flexibility of the EC metric in allowing different cost assignments for different types of errors. However, it does not provide concrete guidelines for practitioners to determine appropriate cost values for their specific applications.

**Strengths And Weaknesses:**

**Strength**
1. This work studies a significant problem and challenges the conventional wisdom regarding the utility of commonly used metrics in decision-making of model development.
2. The preliminaries of the different classification metrics are well-detailed in the main paper.

**Weakness**
1. The empirical evaluation in Section 4 is difficult to follow. It didn't clearly present the evaluation method and clarify what constitutes positive or negative results. The writing in Sections 4.1 and 4.2 reads more like a rough description of the results without providing logical reasoning to explain "highlight the different points we wish to discuss" or "illustrate how the wrong choice of metric can negatively affect development decisions".
2. The presentation of the results is unclear, making it challenging for readers to follow and be convinced by the findings. For instance, interpreting Table 2 is particularly difficult. It might be useful to use bold font type to emphasize which is a row/column that is worth attention.
3. Although the introduction section claims a theoretical contribution, the paper does not present any theorems or lemmas to support its main claim.

---

> ### Author Response · Authors · 2024-08-19
> **Response to review**
>
> We thank the reviewer for the comments and suggestions. Below we respond to the comments.
>
> **Impact of calibration on the EC**
>
> We would like to note that the effectiveness of the EC as a metric does *not* rely on having well-calibrated posteriors. The EC is a metric that assesses the quality of categorical decisions, not the quality of scores. If the threshold used to turn scores into categorical decisions is determined empirically using a development set, as usually done in the machine learning literature for any classification metric, the EC is unaffected by misscalibration. Yet, an advantage of the EC is that it enables the use of Bayes decision theory for decision making in which case its value will reflect how well calibrated the scores are (as well as how discriminative they are). As explained in Section 2.3 this is one of the characteristics that make the EC a better metric compared to alternative metrics like the F1 or the MCC for which there is no corresponding decision theory. The impact of misscalibration on the EC when making Bayes decisions is discussed in Section 4.2.
>
> We do realize that section 2.3 may be giving the impression that having calibrated scores is a requirement for the use of the EC.  We will make sure to clarify this issue in that Section.
>
> **Cost specification**
>
> The problem of how to set the costs highly depends on the application and has been discussed in a number of prior works (eg, Kahneman, 2011, cited in Section 2). As we mention in Section 2, this is a question not for machine learning developers but for subject matter experts. For that reason, it is a discussion that is out of the scope of this paper. We will add some further references on the issue of cost selection for the interested reader.
>
> **Discussion of empirical results**
>
> With respect to weakness number 1, we respectfully disagree. Every subsection in Section 4 has an analysis that goes well beyond a description of results. We do realize that table 2 includes a large number of results and it is hard to parse. The text in that section aims to highlight the most important points in that table, but we agree with the reviewer (as commented in weakness 2) that changes in the font could be used to facilitate comparisons. We will highlight the lines being discussed in each of the paragraphs in that section with colors.
>
> **Theoretical results**
>
> In response to weakness 3, while we are not calling the theoretical results theorems or lemmas, the expression of MCC and F1 as a function of the NEC are novel contributions of this work and support the main claim of this paper that the EC is superior to those alternative metrics.

---

### Review · Reviewer_kbvx · 2024-09-11

**Summary Of Contributions:**

This paper begins by highlighting the limitations of traditional metrics like accuracy, which assumes equal costs for all types of errors. It introduces the EC metric as a generalized form of error rate that considers varying costs for different types of errors. The authors emphasize that EC can be applied to any classification scenario where decisions incur specific costs based on the true class of the sample. The paper argues that, despite its theoretical robustness and flexibility, the EC metric is underutilized in the machine learning community, often being overlooked in favor of simpler, ad-hoc metrics.

The EC metric's primary strength lies in its ability to explicitly incorporate application-specific costs into the evaluation process, providing a more accurate reflection of the true performance of a classification system. This is particularly valuable in domains where some errors are more costly than others, such as in medical diagnostics or financial decision-making. The authors also introduce a normalized version of EC (NEC), which allows for easier interpretation by comparing the performance of a model against a naive baseline that does not consider the input data.

The paper evaluates the effectiveness of EC metrics in various scenarios by means of both theoretical arguments and empirical examples.

**Audience:**

Yes

**Broader Impact Concerns:**

No research ethics concerns were identified.

**Claims And Evidence:**

No

**Requested Changes:**

EC itself is one of the well-known evaluation metrics, and naturally, there have already been various discussions about how to differentiate its use from other metrics. To make the authors' claim that "EC is superior to other evaluation metrics and classification models should fundamentally be evaluated using EC" more convincing, it is necessary to demonstrate that EC is clearly superior to other metrics from various perspectives. For example,
- regarding the cost matrix described in the Weakness section, one approach might be to demonstrate that "setting a cost matrix is easier than using other metrics" or that "EC has a certain degree of robustness against mis-specification of the cost matrix."

In order to unify the numerous evaluation metrics, shouldn't such things be at least experimentally verified?

**Strengths And Weaknesses:**

### Strength
- This paper illustrates the advantages of using Expected Cost (EC) as an evaluation metric for machine learning models in classification problems by contrasting it with other key metrics such as F-beta score and MCC. In particular, I think the part explaining that the other metrics are a special case of EC and how optimizing the other metrics can be interpreted in the context of evaluating the model is well organized.

### Weakness
I understood the authors' argument to be that “EC is superior to other evaluation metrics for classification problems, such as F-beta scores, and that model evaluation in classification problems should essentially be done with EC”. However, EC is not a metric that can be used unconditionally; at least the following two points need to be cleared.

- The outputs of the model should be sufficiently well calibrated and correctly interpreted as posterior probabilities
- Costs for TP, TN, FP and FN should be set appropriately.

The author argues that, for example, in comparison to threshold optimization in other metrics and the setting of the cost matrix in EC, the latter is better (hence EC is superior).
However, this does not seem a very convincing argument. Other evaluation metrics are used in various classification tasks, and there is a wealth of knowledge regarding their characteristics and appropriate use. On the other hand, the specification of the cost matrix is problem-dependent, and it can be said that there is no universally appropriate method for its specification. Additionally, it is unclear how to detect and correct cases where the cost matrix setting significantly deviates from the requirements of the problem.
From this perspective, the authors' claim in this paper that "EC should be used in principle over other metrics" seems to be an overstatement.

---

> ### Author Response · Authors · 2024-09-11
> **Response to review**
>
> Thank you very much for your review.
>
> First, in response to the final question: We believe that the argument that EC is superior to other metrics should not be based on empirical evidence. Metrics should not be selected based on their behavior but based exclusively on the needs of the task. The EC is defined as a direct reflection of the overall cost that a user would incur when using the system and, as such, it is the ultimate classification metric (as also argued in other works, for example, Dyrlan et al$^1$, which will be cited in the revised version of the paper). The main goal of this paper is to show that other standard metrics do not do this. They are not a direct reflection of the cost that the user would incur but, instead, they are affected by other aspects of the system which are unrelated to the goodness of the system. This is shown, for F-score and MCC, by Equations (12) and (14) and then illustrated in the experimental section.
>
> With regards to cost selection, we would like to emphasize that this difficulty is not solved by other metrics. All other classification metrics have an implicit set of costs. So, while the developer does not have to decide on their value, a value is effectively being decided by the metric, as explained in Sections 3.1, 3.2 and illustrated in Section 4.1.1. In fact, as we discuss in those sections, the problem with those metrics is that the costs are not explicit, not selected based on the needs of the task, and not even fixed across systems -- the effective cost that is assigned to a certain error by the F-score differs depending on the system and dataset under evaluation.
>
> In other words, using F-score or MCC gives the user the illusion of having a cost-independent metric. Yet, as we show, this is definitely not the case. The metric is effectively optimizing for a certain set of costs but these costs are implicit and vary with the dataset and system under evaluation. Avoiding the problem of cost selection does not make it go away. This is, we believe, the main argument against the use of such ad-hoc metrics. Again, this is a theoretical argument rather than empirical one.
>
> Finally, note that the use of the EC does not depend on the system being calibrated. One can always use threshold tuning for the EC, as done for other metrics. This is what we do in Figure 1. Yet, if the scores are calibrated the EC offers a possibility that other metrics do not -- using Bayes decision theory to make decisions. Hence, this is not a constraint but, rather, another advantage of the EC.
>
> We look forward to engaging on an interesting discussion on these topics and we thank you again for your review.
>
> [1] K. Dyrland, A. S. Lundervold, and P. Porta Mana. Does the evaluation stand up to evaluation? A first-principle approach to the evaluation of classifiers. arXiv:2302.12006, May 2022.

---

> > ### Comment · Reviewer_kbvx · 2024-11-22
> > **Response to the author's comments**
> >
> > Thank you for your comments, and I apologize for the delayed response.
> >
> > First, I strongly agree with the authors' assertion that evaluation metrics should be aligned with the ultimate KPI of the task, which is not necessarily the case in the current state of machine learning. I also understand the point that EC can provide results more directly connected to KPIs, specifically the user's decision-making process.
> >
> > While I believe it is correct that EC is theoretically desirable as an evaluation metric, I also think that without a clear correspondence to an empirically optimizable objective (which the authors have noted is outside the scope of this paper), it may be challenging to promote its practical adoption. Specifically, in the structure:
> >
> > **task KPI <- expected objective (e.g., EC) <- empirical objective**
> >
> > It is important to simultaneously present what corresponds to the **empirical objective** in order to facilitate the broader adoption of EC. If possible, I would recommend including a discussion on this aspect in the main text.

---

> > > ### Author Response · Authors · 2024-11-25
> > >
> > > Thank you very much for your response and for the recommendation. We will include a longer discussion on the issue of cost-matrix selection and submit a revised version as soon as possible.

---

> > > > ### Author Response · Authors · 2024-11-28
> > > >
> > > > We have uploaded a new version of the paper with a new section that discusses the issue of cost selection. Please, let us know if this discussion addresses your concern.

---

> > > > > ### Comment · Reviewer_kbvx · 2024-12-02
> > > > > **Response to the Authors**
> > > > >
> > > > > Thank you for revising the manuscript. The explicit explanation of the approach to setting the cost matrix has made the use of EC in system evaluation much clearer. Additionally, the inclusion of cost-sensitive examples has effectively highlighted EC as a contrast to traditional metrics. I believe that this revision has improved the quality of the paper.
> > > > >
> > > > > And I apologize if my previous comment was unclear. When I referred to the "empirical objective," I was referring to the section on p.4, specifically the part stating *"using this metric as an objective function for model optimization."* While the paper mentions that *"The use of classification metrics for the purpose of model optimization is outside of the scope of this paper,"* I believe that, practically speaking, optimizing an empirical approximation of EC would be beneficial in terms of ensuring consistency in system optimization.
> > > > > My suggestion was in reference to the statement *"Nevertheless, relaxations of these metrics can be used to turn them into differentiable functions that approximate the original metrics, enabling their use as objective function during optimization."* Including a concrete example to illustrate how this could be achieved would improve clarity and provide more practical insight.

---

> > > > > > ### Author Response · Authors · 2024-12-03
> > > > > >
> > > > > > We apologize for the misunderstanding. Thank you for clarifying.
> > > > > >
> > > > > > We have uploaded a new version where we expanded the explanation on the use of the EC as objective function into a new section 2.5. Please, let us know if this is addresses your concern.

---

> > > > > > > ### Comment · Reviewer_kbvx · 2024-12-04
> > > > > > > **Response to the Authors**
> > > > > > >
> > > > > > > Thank you for promptly addressing my suggestions.
> > > > > > > With the addition of explanations regarding the approach for setting the cost matrix and the methodology for approximating EC during optimization, I believe the paper has gained greater persuasiveness in terms of practical use.

---

### Decision · Action_Editor_6fZ8 · 2025-02-22

**Recommendation:** Accept as is

**Comment:**

There is some debate about whether the work is novel or just a survey of existing ideas. Although the paper builds on well-known theory, its main contribution is the clear framework it provides. The authors show that standard metrics hide important cost trade-ofs that can affect decision-making. This work is not just a review. The paper offers a practical guide on how to select and tune evaluation metrics. Overall, the paper gives useful insights that can help improve future classifier evaluations.

**Audience:**

The findings will interest many readers at TMLR, especially those working on classifier evaluation and cost-sensitive learning. By explaining common metrics using decision theory, the paper makes it easier to understand and improve these metrics for practical applications.

**Claims And Evidence:**

The paper’s claims are supported by clear theory and strong experiments. The authors show that well-known metrics like the F-beta score and M.C..C. can be seen as special cases of the Expected Cost (EC) metric. Even though these ideas come from established decision theory, the paper puts them together in a new way that reveals hidden cost trade-offs. The experimental results on both synthetic and real data add further support.